# Addressing MRSA infection and antibacterial resistance with peptoid polymers

Jiayang Xie[1,4], Min Zhou [1,4], Yuxin Qian[2], Zihao Cong [2], Sheng Chen [2], Wenjing Zhang [2], Weinan Jiang[2], Chengzhi Dai[2], Ning Shao[2], Zhemin Ji[2], Jingcheng Zou [2], Ximian Xiao [2], Longqiang Liu[2], Minzhang Chen[2], Jin Li[3] & Runhui Liu [1,2✉]

Methicillin-Resistant *Staphylococcus aureus* (MRSA) induced infection calls for antibacterial agents that are not prone to antimicrobial resistance. We prepare protease-resistant peptoid polymers with variable C-terminal functional groups using a ring-opening polymerization of *N*-substituted *N*-carboxyanhydrides (NNCA), which can provide peptoid polymers easily from the one-pot synthesis. We study the optimal polymer that displays effective activity against MRSA planktonic and persister cells, effective eradication of highly antibiotic-resistant MRSA biofilms, and potent anti-infectious performance in vivo using the wound infection model, the mouse keratitis model, and the mouse peritonitis model. Peptoid polymers show insusceptibility to antimicrobial resistance, which is a prominent merit of these antimicrobial agents. The low cost, convenient synthesis and structure diversity of peptoid polymers, the superior antimicrobial performance and therapeutic potential in treating MRSA infection altogether imply great potential of peptoid polymers as promising antibacterial agents in treating MRSA infection and alleviating antibiotic resistance.

[1] State Key Laboratory of Bioreactor Engineering, East China University of Science and Technology, 200237 Shanghai, China. [2] Key Laboratory for Ultrafine Materials of Ministry of Education, Frontiers Science Center for Materiobiology and Dynamic Chemistry, Research Center for Biomedical Materials of Ministry of Education, School of Materials Science and Engineering, East China University of Science and Technology, 200237 Shanghai, China. [3] Shanghai Key Laboratory of Orbital Diseases and Ocular Oncology, Department of Ophthalmology, Ninth People's Hospital, Shanghai Jiao Tong University School of Medicine, 200011 Shanghai, China. [4] These authors contributed equally: Jiayang Xie, Min Zhou. ✉email: rliu@ecust.edu.cn

Methicillin-resistant *Staphylococcus aureus* (MRSA) is frequently encountered in hospitals and communities worldwide, with high morbidity and mortality, and is a formidable threat to human health[1,2]. The sustained emergence and rapid spread of MRSA as well as the lack of new antibiotics imply the post-antibiotic era and an urgent call for novel antimicrobial agents[3,4]. Host-defense peptides (HDPs) have been studied as promising potential therapeutic alternatives because of their broad-spectrum antimicrobial activity and low susceptibility to antimicrobial resistance[5–9]. However, HDPs' application is limited due to their innate shortcomings, including their low stability in the presence of protease, high cost, and time-consuming step-by-step synthesis[10,11]. Therefore, peptide mimetics[12–19], as well as polymeric mimetics, have been studied to retain the favorable antimicrobial functions of HDPs and, in the meantime, to address HDPs' shortcomings[20–38].

Peptoids have been studied as a class of peptide mimetics, with their structure differing from peptides only in that peptoids' side chains are attached to the backbone amide nitrogen rather than to the α-carbon[39], which enables peptoids' excellent stability in the presence of protease[40,41]. Peptoids have been explored as antibacterial mimics of HDPs in precedent literatures[42–47], but with a very little report for in vivo demonstration[48,49]. Moreover, these HDP mimicking peptoids were mostly prepared from time-consuming step-by-step solid-phase synthesis[50].

In this study, we design and synthesize a series of peptoid polymers, poly-*N*-aminoethylglycine (poly-Naeg), via the convenient one-pot ring-opening polymerization (ROP) of α-amino acid *N*-substituted *N*-carboxyanhydrides (α-NNCAs) (Fig. 1a). The optimal peptoid polymer displays superior antibacterial properties: with effective activities against MRSA, persister cell killing, effective eradication of MRSA biofilms, and in vivo anti-infectious effectiveness in a mouse wound model, a mouse keratitis model, and a mouse peritonitis model induced by MRSA (Fig. 1b). It is noteworthy that bacteria are unable to acquire resistance against the peptoid polymer owing to the antibacterial mechanism, including the generation of reactive oxygen species (ROS).

## Results

**Synthesis of peptoid polymers and in vitro antibacterial study**. A previous study indicated that the biological function of antimicrobial polymers can be tuned by their terminal functional groups[51], which inspired us to explore peptoid polymers with hydrophilic group $PEG_4$ or hydrophobic groups, variable aromatic groups and variable alkyl chains. The *N*-substituted α-NNCA monomer, $N^\beta$-Cbz-aminoethyl-NNCA (Supplementary Figs. 1–7), was polymerized using primary amines as the initiators followed by an acidic deprotection step to give 20 mer peptoid polymers[52–54], poly-Naeg, with variable C-terminal functional groups (Poly1 to Poly7) and narrow dispersities (Đ = 1.08–1.12, dispersity (Đ) is a measure of the dispersion of macromolecular species in a sample of polymer, i.e. a measurement of the heterogeneity of sizes of molecules or particles in a mixture, calculated from the ratio of $\bar{M}_w$ to $\bar{M}_n$) (Fig. 1a and Supplementary Figs. 8–14, 19–25). We examined the activity of these poly-Naeg against multiple Gram-positive bacteria, including five strains of MRSA (*S. aureus* Mu50, *S. aureus* Newman, *S. aureus* USA400, *S. aureus* USA300 LAC, and *S. aureus* USA300) and three drug-sensitive species (*S. aureus* ATCC6538, *S. epidermidis* 49134, and *B. subtilis* BR-151). All poly-Naeg showed effective activity against these bacteria within our test, with thiol terminated Poly1 performing slightly better and having MIC (the minimum inhibitory concentration) values and MBC (the minimum bactericidal concentration) values in the range of 3.13–12.5 μg/mL and 3.13–

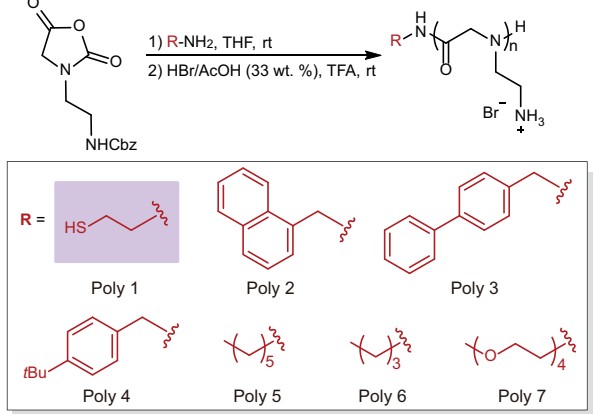

**a** Synthesis of Poly-α-peptoids with various C-terminals

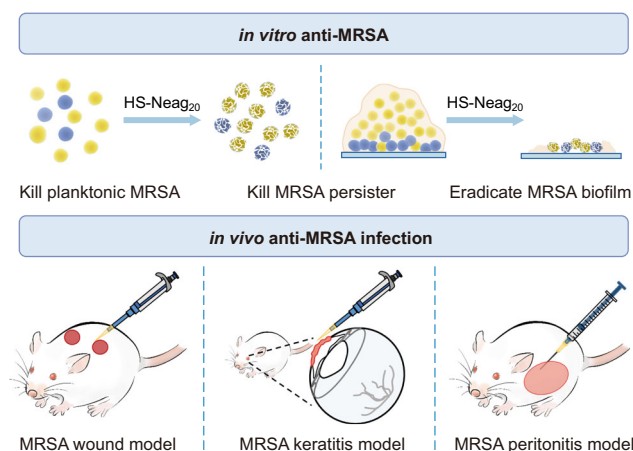

**b** *In vitro* and *in vivo* antibacterial of Poly-α-peptoids

**Fig. 1 Design and synthesis of peptoid polymers with anti-MRSA activity. a** Synthesis of poly-α-peptoids bearing variable C-terminal functional groups. *n* represents the average number of repeating units within the peptoid polymer chain, that is, the degree of polymerization. **b** Schematic illustration on the antibacterial performance of poly-Naeg, in vitro and in vivo.

25 μg/mL, respectively (Fig. 2a). We then took the thiol terminated Poly1 as the selected compound for more detailed studies.

We prepared thiol terminated poly-Naeg at variable chain lengths by controlling the ratio of initial monomer/initiator (DP = 5, 10, 20, 40; Fig. 2b and Supplementary Figs. 15–18, 26–29) and evaluated their activities against *S. aureus* ATCC6538. These poly-Naeg ($HS(Naeg)_5$, $HS(Naeg)_{10}$, $HS(Naeg)_{20}$, and $HS(Naeg)_{40}$) showed chain length-dependent antibacterial activity, the longer polymers at 20 and 40 mer showing a better activity with MIC of 12.5 μg/mL (Fig. 2c). Though both $HS(Naeg)_{20}$ and $HS(Naeg)_{40}$ showed low hemolysis and cytotoxicity, $HS(Naeg)_{20}$ was superior to $HS(Naeg)_{40}$ in having $HC_{50}$ (the minimum concentration to cause 50% hemolysis) value over 10000 μg/mL, $IC_{50}$ (the minimum concentration to caused 50% reduction in cell viability) value of 400 μg/mL, and high antibacterial selectivity index ($HC_{50}/MIC > 800$ and $IC_{50}/MIC = 32$) (Fig. 2d–g). The optimal peptoid polymer $HS(Naeg)_{20}$ was further tested on other different clinically isolated Gram-positive strains. The polymer is active against these Gram-positive bacteria with MIC values in the range of 1.56–25 μg/mL. In sharp contrast, most of the clinically isolated strains we tested were drug-resistant or even multiple drug-resistant (Table 1). It's noteworthy that the MIC and MBC values of the peptoid polymer keep almost constant when tested in the presence of serum or

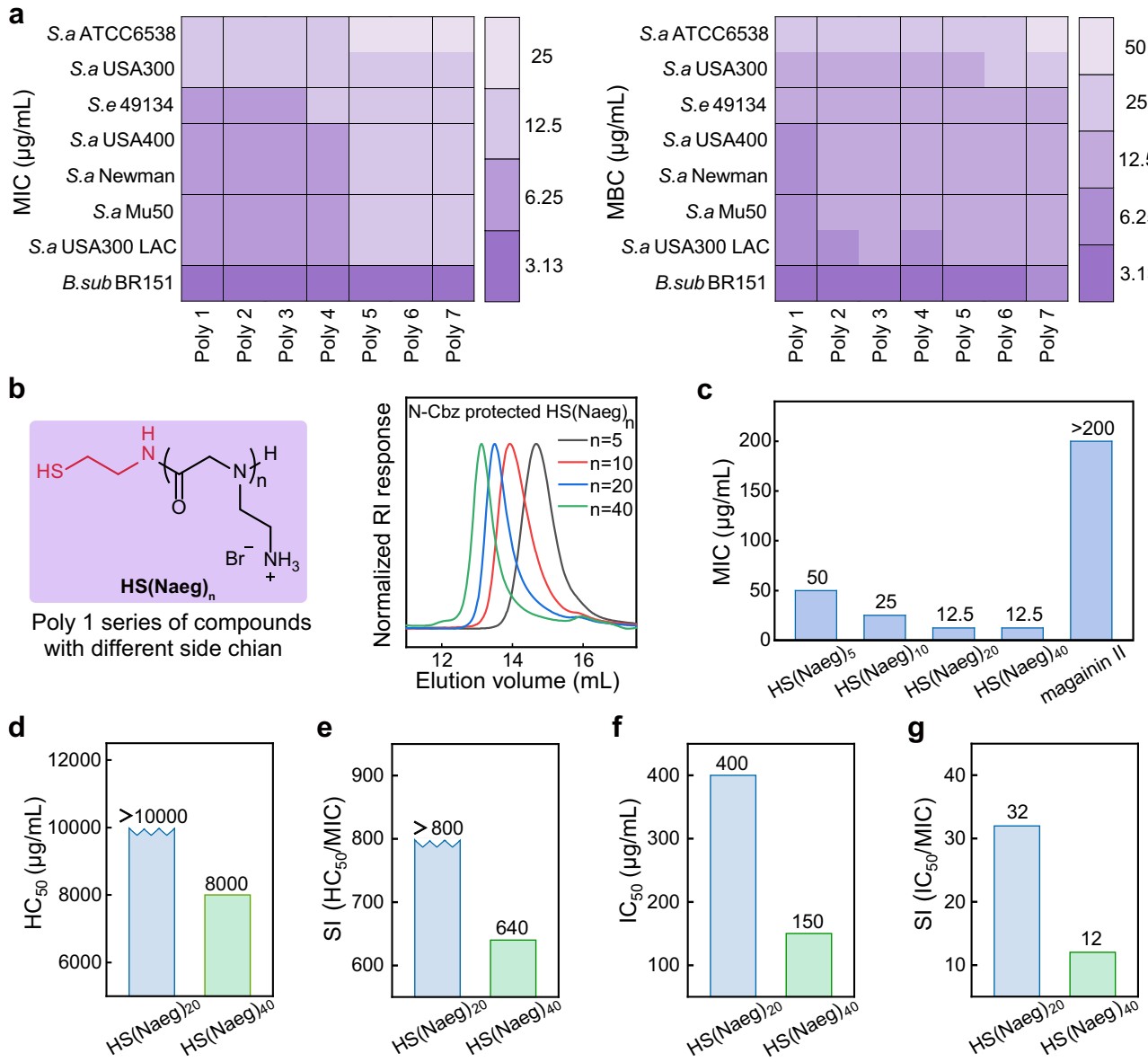

**Fig. 2 Poly-α-peptoids displayed an excellent in vitro antibacterial profile. a** Antibacterial activity of Poly-α-peptoids. **b** Chemical structure of HS(Naeg)$_n$ and GPC characterization of *N*-Cbz protected HS(Naeg)$_n$ ($n = 5, 10, 20, 40$). **c** MIC values of HS(Naeg)$_n$, at variable chain length, against *S. aureus* ATCC6538 using magainin II for comparison. **d** Hemolysis of HS(Naeg)$_{20}$ and HS(Naeg)$_{40}$ on human red blood cells. **e** Selectivity index calculated from HC$_{50}$/MIC. **f** Cytotoxicity of HS(Naeg)$_{20}$ and HS(Naeg)$_{40}$ on fibroblast cells. **g** Selectivity index calculated from IC$_{50}$/MIC.

salts used to simulate the physiological environment (Supplementary Fig. 34 and Supplementary Table 1).

**Antibacterial mechanism.** To figure out the antibacterial mechanism of poly-Naeg, we did cytoplasmic membrane depolarization test using DiSC3(5) dye as the indicator and found that HS(Naeg)$_{20}$ only displayed a moderate membrane depolarization effect on *S. aureus* (Fig. 3a). To get more detailed mechanistic information using time-laps fluorescent confocal imaging, we prepared a morpholino-naphthalimide dye functionalized poly-Naeg, dye–(Naeg)$_{20}$, via *N*-(3-aminopropyl)-4-morpholin-1,8-naphthalimide initiated NNCA polymerization followed by deprotection under acidic conditions (Supplementary Figs. 30–33). Incubating *S. aureus* with dye–(Naeg)$_{20}$ (green fluorescence) at $1 \times$ MBC concentration and propidium iodide (PI, red fluorescence), we observed that the polymer entered into

cytoplasm directly from ~400 s without strong enrichment on the cell membrane and without PI signal (Fig. 3b, c), which echoed the weak membrane depolarization conclusion above and implied an antimicrobial mechanism without strong direct interactions between the polymer and bacteria membrane. We hypothesize that peptoid polymer penetrates membrane possibly through direct translocation like some cationic antimicrobial peptides reported in precedent literature: the interaction of the positively charged polymer with negatively charged components of bacteria membrane destabilizes the membrane bilayer, creating a trans-membrane channel, thereby allowing the peptoid polymer to enter the bacterial cells without the membrane lysis[55]. About 300 s after the peptide polymer enriched in the cytoplasm, PI got into the cytoplasm and indicated the damage of bacteria membrane (Fig. 3b, c), which was supported by the front view of the Z-stacked image confirming colocalization of polymer with PI within bacteria (Fig. 3d).

**Table 1 Antibacterial activities against clinically isolated Gram-positive strains.**

| Strains | MIC (µg/mL) | | | | |
|---|---|---|---|---|---|
| | HS(Naeg)$_{20}$ | Vancomycin | Methicillin | Norfloxacin | Ampicillin |
| S. aureus 2904 | 6.25 | 0.39 | 6.25 | 200 | 25 |
| S. aureus 2802 | 6.25 | 0.39 | 6.25 | 200 | 25 |
| S. aureus 2902 | 6.25 | 0.39 | 12.5 | 25 | 25 |
| S. aureus 2202 | 6.25 | 0.78 | 3.13 | 50 | 1.56 |
| S. haemolyticus 1303 | 1.56 | 0.78 | >200 | 25 | 200 |
| S. haemolyticus 1107 | 25 | 0.78 | >200 | 100 | >200 |
| S. haemolyticus 0202 | 3.13 | 1.56 | 200 | 100 | 200 |
| S. epidermidis 9397 | 3.13 | 0.78 | 1.56 | 0.2 | 0.39 |
| S. epidermidis 0501 | 3.13 | 0.78 | 12.5 | 0.2 | 6.25 |
| E. faecium 2504 | 6.25 | 0.78 | >200 | 200 | 100 |
| E. faecium 0610 | 25 | 0.78 | >200 | 200 | >200 |
| E. faecium 1205 | 25 | 0.39 | >200 | 100 | >200 |
| E. faecium 0502 | 12.5 | >200 | >200 | 100 | >200 |
| E. faecium 0609 | 25 | 50 | >200 | >200 | >200 |
| E. faecalis 2305 | 25 | 0.78 | >200 | 25 | 12.5 |
| E. faecalis 0609 | 25 | 0.39 | 50 | 1.56 | 3.13 |
| S. agalactiae 0613 | 25 | 0.78 | 1.56 | 25 | 0.39 |
| L. monocytogenes 1001 | 12.5 | 0.39 | 25 | 1.56 | 1.56 |

We also found that the peptoid polymer HS(Naeg)$_{20}$ bound to DNA strongly even at a low N:P ratio of 1:1 (Fig. 3e), which may inhibit cellular functions and cause bacterial death. DNA binding may also lead to the production of ROS and kill bacteria by damaging bacteria membrane as reported in literatures[56,57]. Using 2',7'-dichlorofluorescin diacetate as the ROS indicator, we found the ROS level within S. aureus increased around 5.5-fold after the bacteria were incubated with HS(Naeg)$_{20}$ for 30 min. The addition of 10 mM ROS inhibitor N-acetyl-L-cysteine (NAC) suppressed intracellular ROS to normal cell level (Fig. 3f), which was accompanied by the loss of activity against S. aureus (MBC > 1000 µg/mL) for HS(Naeg)$_{20}$ (Fig. 3g). The fact that the antioxidant reagent NAC blocks bacterial killing and that peptoid polymer kills bacteria much faster than do DNA-targeting antibacterial agents[58–60] suggest that bacterial killing of peptoid is associated with the generation of the high level of ROS, rather than just interaction or damage on DNA (Fig. 3g and Supplementary Fig. 40). These studies suggested the probable complex antibacterial mechanism of poly-Naeg including the generation of the high level of ROS and DNA binding[7,61,62]. The generation of the high levels of ROS could kill bacteria by damaging bacterial membrane which was supported by the analysis using a scanning electron microscope (SEM) and transmission electron microscopy (TEM). SEM characterization on HS(Naeg)$_{20}$ (1 × MBC)-treated S. aureus showed obviously damaged bacterial membrane (Fig. 3h). TEM characterization on HS(Naeg)$_{20}$-treated S. aureus showed cytoplasm membrane damage and loss of cytoplasmic content (Fig. 3h).

**Insusceptibility to antibacterial resistance and fast bacterial killing.** Antibacterial resistance test on HS(Naeg)$_{20}$, the best antibacterial peptoid polymer with this study, showed that S. aureus did not acquire resistance upon polymer even after the bacteria was treated continuously with HS(Naeg)$_{20}$ at a sublethal dose for 834 passages; in sharp contrast, the MBC of norfloxacin increased by 1024 times after S. aureus was treated with norfloxacin for 375 passages (Fig. 4a). The time-kill kinetic study showed that HS(Naeg)$_{20}$ achieved about 2.7-log reduction of S. aureus within 60 min at a concentration of 1 × MBC. In sharp contrast, vancomycin caused only about 0.3-log reduction of the bacteria even after 4 h of treatment (Fig. 4b). Compared with

conventional antibiotics, the fast bacterial killing is one of the advantages of our peptoid polymers, which is very important in the treatment of sepsis and other situations where there is an urgent need to kill bacteria.

**Eradication of persister bacteria and biofilm.** The ability of HS(Naeg)$_{20}$ to kill persister cells was evaluated, and the result showed that the peptoid polymer can effectively kill persister cells that is known to be a key reason for the high antimicrobial resistance associated with mature biofilms (Fig. 4c). The ability of HS(Naeg)$_{20}$ to kill persister cells encouraged us to explore this peptoid polymer for its activity against S. aureus biofilms, a formidable threat to human health due to the frequently encountered strong drug resistance[63,64]. HS(Naeg)$_{20}$ efficiently inhibited the formation of S. aureus biofilms at a concentration as low as 1 × MIC; for two antibiotic controls, vancomycin and norfloxacin, required a concentration of 2 × MIC and 64 × MIC respectively to inhibit the formation of S. aureus biofilm (Fig. 4d). For the even more challenging mature biofilms of S. aureus, HS(Naeg)$_{20}$ eradicated the biofilms efficiently at a concentration of 8 × MIC; whereas, vancomycin and norfloxacin could not eradicate mature biofilms effectively even at a concentration up to 1024 × MIC and 2048 × MIC, respectively (Fig. 4e).

**The in vivo anti-infectious efficacy of HS(Naeg)$_{20}$.** Encouraged by the superior antibacterial performance of the peptoid polymers, we continued to evaluate the in vivo antibacterial efficacy of HS(Naeg)$_{20}$ in the mouse full-thickness wound model and the mouse keratitis model, including vancomycin and saline as the positive and negative control, respectively. In the wound infection model, MRSA suspension was applied to the wound and infected for 24 h followed by topical treatments with peptoid polymer. Compared with the saline group, HS(Naeg)$_{20}$ treatment led to a significant reduction in the bacterial burden on the wound (~2.0-log reduction), which is better than vancomycin treatment (~1.2-log reduction) (Fig. 5a). Besides, we also used the same model in immunosuppressed mice to explore the in vivo activity of HS(Naeg)$_{20}$ against other Gram-positive bacteria such as S. epidermidis and S. haemolyticus. It was found that the polymer performed the same as or even superior to did vancomycin for the infection caused by S. epidermidis and S. haemolyticus,

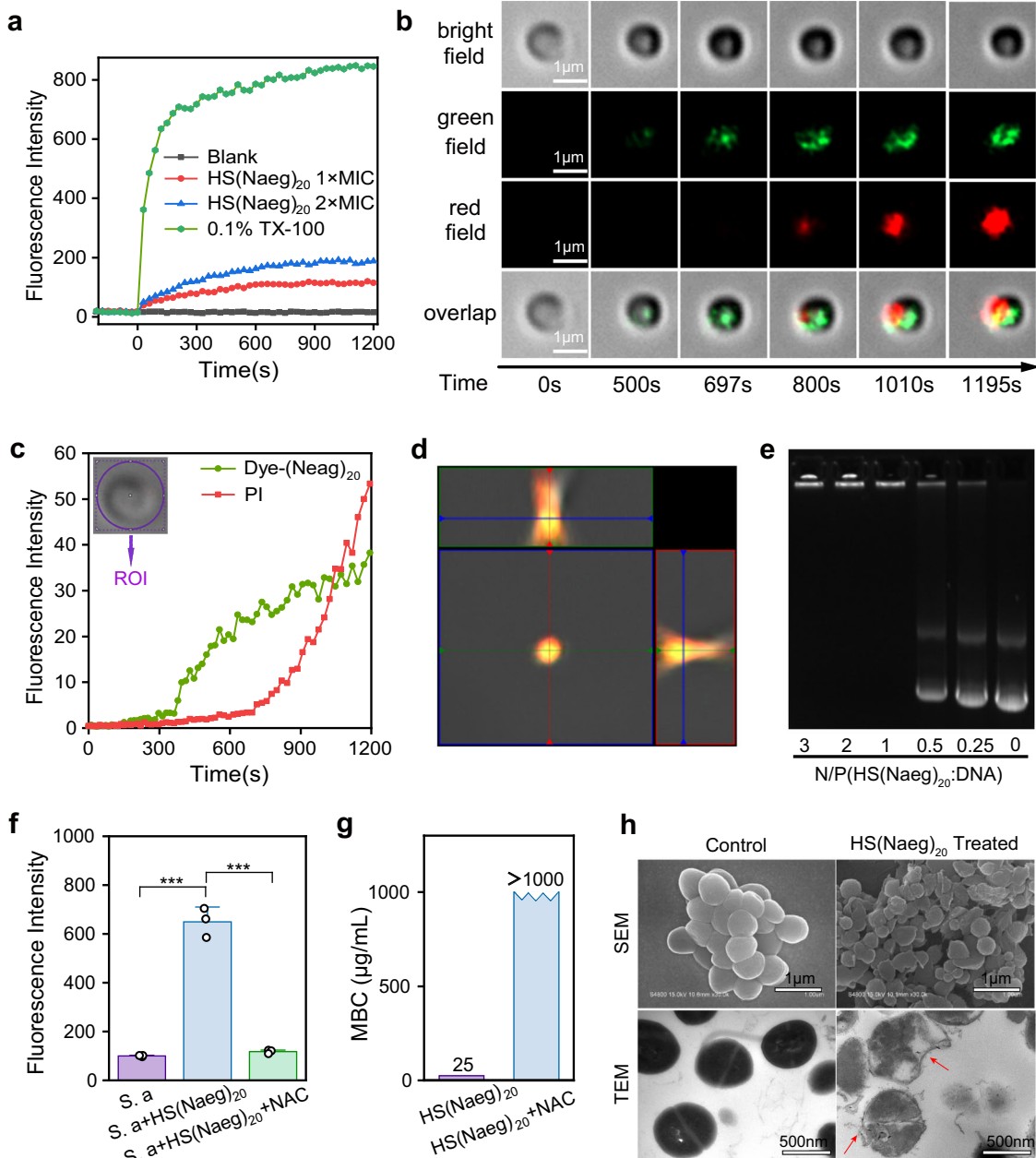

**Fig. 3 Antibacterial mechanism study of poly-Naeg. a** Cytoplasmic membrane depolarization by HS(Naeg)$_{20}$ at 1×MIC and 2×MIC concentrations.
**b** Time-laps confocal fluorescence imaging on the interaction between *S. aureus* and dye–(Naeg)$_{20}$ at a concentration of 1×MBC, in the presence of PI.
**c** Fluorescence intensity versus time in green and red channels in ROI. **d** Ortho view of Z-stack images in (**b**). **e** The electrophoretic mobility shift assay of plasmid DNA and the mixture of plasmid DNA-complexes at different ratios of N:P (HS(Naeg)$_{20}$:DNA). **f** Fluorescence intensity produced by *S. aureus* treated with PBS buffer, HS(Naeg)$_{20}$ (1 × MBC) and the combination of HS(Naeg)$_{20}$ (1 × MBC) and NAC (10 mM), in the presence of 2′,7′-dichlorofluorescin diacetate ($n = 3$ per group). Data are presented as mean ± SD. ***$P < 0.001$ (Student's $t$ test). **g** The MBC of HS(Naeg)$_{20}$ against *S. aureus* in the presence or absence of NAC (10 mM). **h** SEM and TEM characterization respectively on *S. aureus* cells with and without HS(Naeg)$_{20}$ treatment at a concentration of 1 × MBC.

respectively (Fig. 5b, c). We also evaluated the in vivo antibacterial activity of HS(Naeg)$_{20}$ against biofilm in a MRSA keratitis model. In this model, the polymer reduced the bacterial load of each eye of the mouse by 2.5-log, while vancomycin was ineffective (Fig. 5d).

Furthermore, the mouse peritonitis model was employed to evaluate the therapeutic potential of HS(Naeg)$_{20}$ on systemic infection. Mice were infected by intraperitoneal injection of MRSA at a lethal dose (all untreated mice died within 12 h). Intraperitoneal injection of a single dose of polymer (20 mg/kg)

can substantially increase the survival rate of infected mice (5/6 mice survived) and reduce the bacterial load in major organs, blood and peritoneal fluid, making it comparable to vancomycin in antibacterial efficacy (Fig. 6a, d). We also examined the typical pathological changes of the organs, and found that these changes were alleviated if with peptoid polymer treatment (Supplementary Fig. 38). Likewise, we examined the effectiveness of HS(Naeg)$_{20}$ to treat peritonitis infections induced by *S. epidermidis* and *S. haemolyticus* in immunosuppressed mice. In the peritonitis model of *S. epidermidis* (66.7% death in saline treatment),

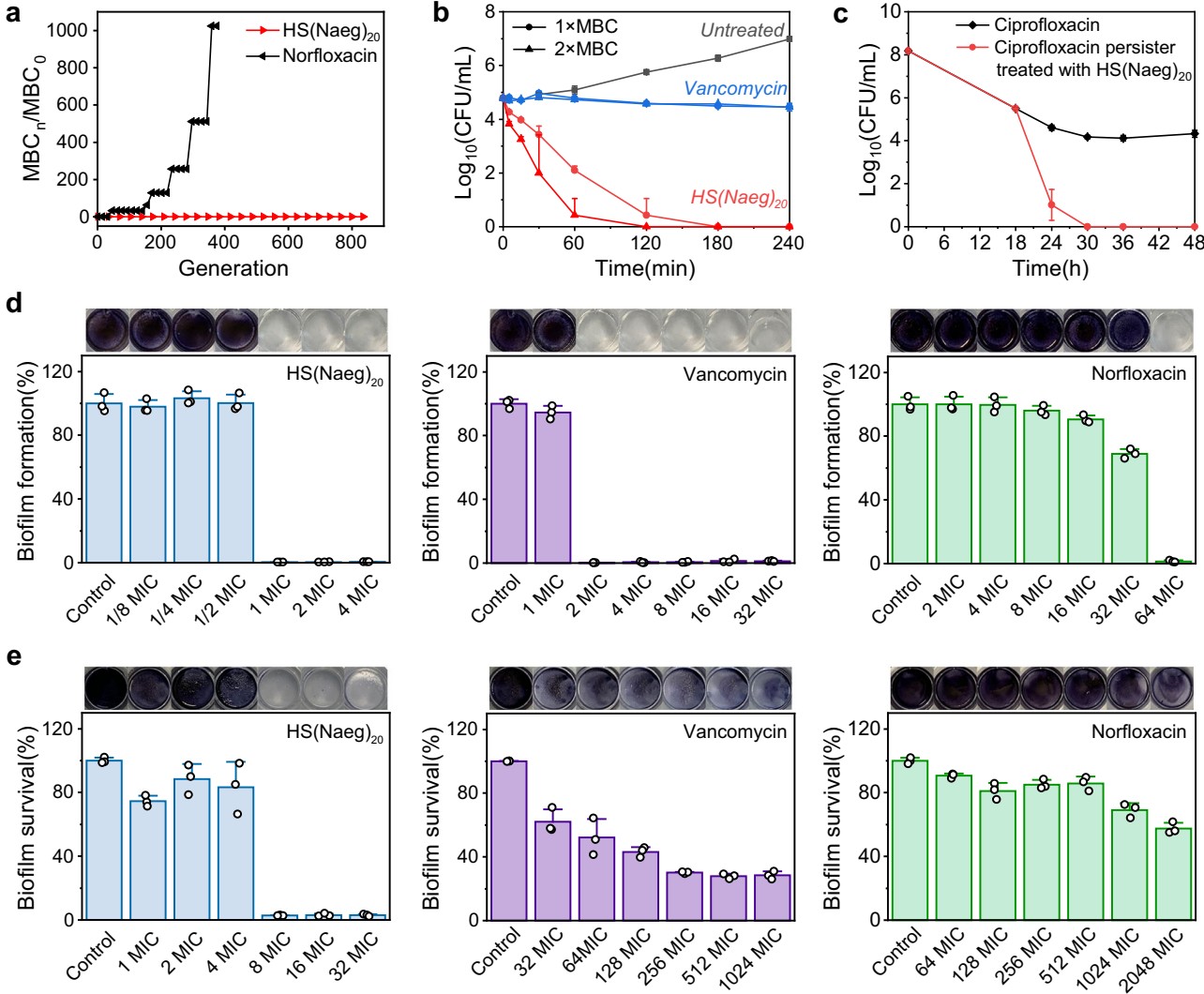

**Fig. 4 Selected HS(Naeg)$_{20}$ displayed insusceptibility to antibacterial resistance, fast bacterial killing, ability to kill persister cells, and effective activity upon biofilms. a** Antibacterial resistance test on HS(Naeg)$_{20}$ and norfloxacin against *S. aureus* ATCC6538. **b** Bacterial killing kinetics of HS(Naeg)$_{20}$ and vancomycin against *S. aureus* ATCC6538 at 1 × MBC and 2 × MBC concentration. **c** Killing kinetics of HS(Naeg)$_{20}$ against persister cells generated by high-concentration ciprofloxacin treatment. **d** The inhibitory effect of HS(Naeg)$_{20}$, norfloxacin, and vancomycin against MRSA biofilm formation. **e** The ability of HS(Naeg)$_{20}$, norfloxacin and vancomycin to eradicate mature MRSA biofilm. $n = 3$ per group, data are presented as mean ± SD.

polymer treatment resulted in 100% survival of the mice (6/6 mice survived), which was better than vancomycin (4/6 mice survived) (Fig. 6b). When the infection was caused by *S. haemolyticus* (all untreated mice died within 60 h), both polymer and vancomycin treatment achieved 100% survival of the mice (Fig. 6c). In addition, the peptoid polymer was equally as effective as vancomycin in reducing the bacterial burden in organs, blood, and peritoneal fluid in both two models (Fig. 6e, f). Finally, in vivo toxicity studies showed that there were no obvious differences in clinically important biomarkers of liver and kidney and no apparent tissue damage at 2 and 7 days post treatment with HS(Naeg)$_{20}$ compared with the blank control group (Fig. 6g and Supplementary Fig. 39). It is worth mentioning that numerous peptidomimetic polymers with antibacterial properties have been reported in precedent studies, but most of them were proof-of-concept demonstrations of in vitro antibacterial activity and simple in vivo studies. The low in vivo toxicity and high in vivo antibacterial efficacy in multiple animal models demonstrated that our optimal peptoid polymer is a promising candidate for therapeutic agents.

## Discussion

Peptoids have been explored as promising antimicrobial mimics of host-defense peptides to address the formidable challenge of antibiotic-resistant bacterial infections. However, the popular step-by-step solid-phase synthesis of peptoids is high cost, time-consuming, and difficult for large-scale synthesis, which hindered the application of antimicrobial peptides. Hereby, we report antibacterial peptoid polymers from the one-pot ring-opening polymerization of NNCA, with low cost, fast, and easy large-scale synthesis. These peptoid polymers displayed effective activities against the "super bugs" MRSA on both planktonic and persister cells and effectively inhibited MRSA biofilm formation. Even for the challenging mature MRSA biofilms that are highly resistant to antibiotics, such as vancomycin and norfloxacin, the selected peptoid polymer effectively eradicated the biofilms and killed MRSA inside. The peptoid polymer also displayed effectively in vivo anti-infectious function in three MRSA-infected animal models, as well as *S. epidermidis* and *S. haemolyticus* infection models. Moreover, after repeated use of the antibacterial peptoid polymer at a sublethal dose for 834 passages, *S. aureus* did not

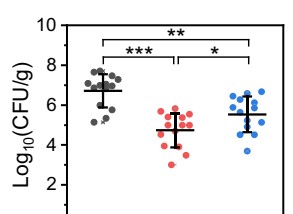

**a** MRSA wound model

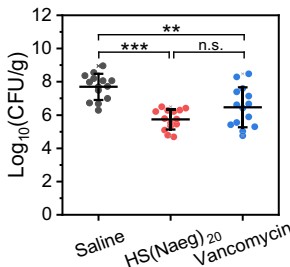

**b** *S. epidermidis* wound model

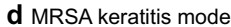

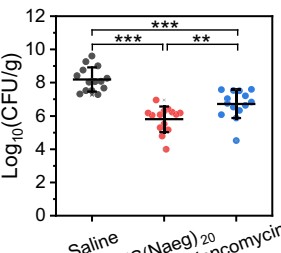

**c** *S. haemolyticus* wound model

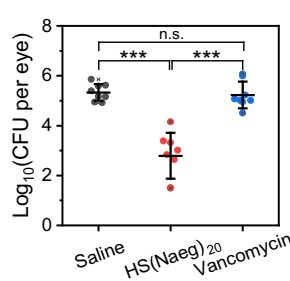

**d** MRSA keratitis model

**Fig. 5 HS(Naeg)₂₀ displayed in vivo antibacterial efficacy in the mouse full-thickness wound model and the mouse keratitis model. a–c** In the mouse wound model, bacterial suspension was applied to the wound and infected for 24 h followed by topical treatments. For infections caused by *S. epidermidis* and *S. haemolyticus*, immunosuppressed mice were used. **a** CFU of MRSA in wound treated with saline, HS(Naeg)₂₀, or vancomycin, all at 1.56 mg/mL ($n = 7$ mice per group). **b** CFU of *S. epidermidis* in wound treated with saline, HS(Naeg)₂₀, or vancomycin, all at 3.13 mg/mL ($n = 7$ mice per group). **c** CFU of *S. haemolyticus* in wound treated with saline, HS(Naeg)₂₀, or vancomycin, all at 3.13 mg/mL ($n = 7$ mice per group). **d** In the mouse keratitis model, contact lens with MRSA biofilm was placed on the injured cornea surface and infected for 12 h followed by topical treatments. CFU of MRSA in cornea treated with saline, HS(Naeg)₂₀, or vancomycin, all at 1.56 mg/mL ($n = 4$ mice per group). Data are presented as mean ± SD. *$P < 0.05$, **$P < 0.01$; ***$P < 0.001$; n.s. (not significant) represents $P > 0.05$ (Student's *t* test).

acquire resistance. These observations were consistent with the bacterial killing mechanism of the peptoid polymer. This highly favorable insusceptibility to antibacterial resistance, in addition to the structural versatility and aforementioned superior in vivo antibacterial performance of peptoid polymers, imply a great potential of peptoid polymers in developing antimicrobial agents and antimicrobial materials.

## Methods

**Measurements**. Nuclear magnetic resonance (NMR) spectra were collected on an AVANCE III 400 spectrometer at 400 MHz or an Ascend 600 spectrometer at 600 MHz using CDCl₃ or D₂O as solvents. Chemical shifts were referenced to the resonance for residual protonated deuterated solvent. High-resolution electrospray ionization time-of-flight mass spectrometry (HRESI-MS) was collected on a Waters XEVO G2 TOF mass spectrometer and high-resolution electron ionization time-of-flight mass spectrometry (HREI-MS) was collected on a Waters GCT Premier mass spectrometer. Gel permeation chromatography (GPC) was performed on a Waters GPC instrument equipped with an isocratic HPLC pump (Waters 1515), a Brookhaven BI-MwA multi-angle light scattering detector, and a refractive index detector (Waters 2414). N, N-dimethylformamide (DMF, supplemented with 10 μM LiBr) was used as the mobile phase at a flow rate of 1 mL/min at 50 °C. The GPC was equipped by a Tosoh TSKgel Alpha-2500 column (particle size 7 μm) and a Tosoh TSKgel Alpha-3000 column (particle size 7 μm) linked in series. Relative number-average molecular weight, degree of polymerization (DP), and dispersity index (Đ) were calculated from a calibration curve using polymethylmethacrylate (PMMA) as standards. SEM images were characterized on a Hitachi S-4800 field emission SEM and TEM images were characterized on a FEI Tecnai Spirit TEM. Time-laps fluorescent confocal imaging was performed using

ZEISS LSM 880 with Airyscan system in ZEISS Microscopy Shanghai Customer Centers.

**Synthesis of *N*-[2-(benzyloxycarbonylamino)ethyl]glycine ethyl ester (compound 1)[65]**. 1-(Benzyloxycarbonylamino)-2-aminoethane (3.9 g, 20.0 mmol) was dissolved in 150 mL dichloromethane (CH₂Cl₂), followed by the addition of ethyl 2-bromoacetate (3.3 g, 20.0 mmol) and triethylamine (2.2 g, 22.0 mmol) sequentially and the reaction was stirred at 55 °C for 12 h. After cooling to room temperature (rt), the reaction mixture was washed with deionization (DI) water three times (3 × 150 mL), and then the organic phase was dried over anhydrous MgSO₄. After removing the solvent under the vacuum, the crude product was purified by silica gel column chromatography to obtain compound 1 as a yellow oil (3.0 g, 53.5% yield). ¹H NMR (400 MHz, CDCl₃): δ 7.33–7.28 (m, 5H), 5.52 (br, 1H), 5.07 (s, 2H), 4.15 (q, *J* = 7.2 Hz, 2H), 3.35 (s, 2H), 3.25 (q, *J* = 5.6 Hz, 2H), 2.73 (t, *J* = 5.8 Hz, 2H), 1.87 (s, 1H), 1.24 (t, *J* = 7.2 Hz, 3H). HREI-MS: *m/z* calculated for C₁₄H₂₀N₂O₄ [M]⁺: 280.1423; found: 280.1419.

**Synthesis of *N*-[2-(benzyloxycarbonylamino)ethyl]-*N*-[(tert-butoxy)carbonyl] glycine (compound 2)**. The synthesis of compound 2 was conducted by following a precedent procedure with modifications[66]. Compound 1 (2.8 g, 10.0 mmol) was dissolved in 100 mL methanol (MeOH), followed by the addition of di-tert-butyl dicarbonate (4.4 g, 20.0 mmol) to the reaction. The reaction mixture was heated to 60 °C and stirred overnight. After removing the solvent under the vacuum, an intermediate was obtained and used directly for the next reaction. The intermediate was dissolved in the mixed solvent of MeOH (60 mL) and tetrahydrofuran (THF, 20 mL), followed by the addition of 1.0 M NaOH solution to the mixture slowly. After stirring at rt for 5 h, the mixture was adjusted to pH 7 with 1.0 M HCl solution, and then the organic solvent was removed under vacuum. The residual aqueous phase was acidified by dropwise addition of 1.0 M HCl solution at 0 °C until pH 3–4. The mixture was extracted with ethyl acetate (EtOAc) three times (3 × 50 mL), and then the combined organic phase was dried over anhydrous MgSO₄ and concentrated under a vacuum. The crude product was purified by silica gel column chromatography to obtain compound 2 as a white solid (3.0 g, 85.1% yield over two steps). ¹H NMR (400 MHz, CDCl₃): δ 7.35–7.31 (m, 5H), 5.51 (br, 1H), 5.11 (s, 2H), 3.96 (s, 1H), 3.91 (s, 1H), 3.42-3.31 (m, 4H), 1.42 (d, *J* = 6 Hz, 9H). HRESI-MS: *m/z* calculated for C₁₇H₂₄N₂NaO₆ [M + Na]⁺: 375.1532; found: 375.1531.

**Synthesis of monomer *N*-[2-(benzyloxycarbonylamino)ethyl]glycine-*N*-carboxyanhybride (Nᵝ-Cbz-aminoethyl-NNCA)**. Compound 2 (2.0 g, 5.7 mmol) was dissolved in 100 mL anhydrous CH₂Cl₂ under a nitrogen atmosphere, followed by dropwise addition of phosphorus tribromide (1.5 g, 5.7 mmol) to the mixture at 0 °C. The reaction mixture was then warmed up to rt and stirred for 1 h. The reaction mixture was washed once with DI water (100 mL) quickly, and then the collected organic phase was dried over anhydrous MgSO₄ and concentrated under a vacuum. The crude product was recrystallized from a mixture of dichloromethane/hexane (v/v = 1:3) to obtain Nᵝ-Cbz-aminoethyl-NNCA as a white needle-like crystal (0.8 g, 50.7% yield). ¹H NMR (600 MHz, CDCl₃) δ 7.36-7.30 (m, 5H), 5.34 (br, 1H), 5.06 (s, 2H), 4.16 (s, 2H), 3.46 (t, *J* = 5.7 Hz, 2H), 3.38 (t, *J* = 6 Hz, 2H). ¹³C NMR (150 MHz, CDCl₃): δ 165.58, 157.09, 153.02, 136.26, 128.71, 128.42, 128.18, 67.19, 49.34, 43.82, 38.32. HRESI-MS: *m/z* calculated for C₁₃H₁₄N₂NaO₅ [M + Na]⁺: 301.0800; found: 301.0801.

**Synthesis of *N*-Cbz protected Poly(Naeg)₂₀**. All polymerizations were carried out in a glove box under a nitrogen atmosphere. The C-terminal functional group of poly(Naeg)₂₀ was introduced by using different amine initiators. The monomer Nᵝ-Cbz-aminoethyl-NNCA and a primary amine reagent were dissolved in anhydrous THF to a final concentration of 1.0 M and 0.5 M, respectively. Take the polymerization of hexylamine functionalized poly(Naeg)₂₀ for example, the *N*-Cbz protected polymer was synthesized from a mixture of Nᵝ-Cbz-aminoethyl-NNCA solution (0.18 mL) and 1-hexylamine solution (0.012 mL). The polymerization proceeded at rt until all monomers were completely consumed. Then the reaction mixture was diluted with 1 mL MeOH followed by the addition of cold petroleum ether (PE, 45 mL) to the mixture to precipitate out the polymer as white flocculent sediment. The polymer precipitate was collected by centrifugation and removal of the supernatant. After repeating three times of the above dissolution–precipitation process, the purified polymer was obtained. The *N*-Cbz protected polymers, with various C-terminal functional groups, were characterized by GPC using DMF as the mobile phase at a flow rate of 1 mL/min.

**Synthesis of *N*-Cbz protected HS(Naeg)ₙ at a variable chain length**. The *N*-Cbz protected HS(Naeg)ₙ at variable chain length were prepared by controlling the ratio of monomer/initiator. The monomer Nᵝ-Cbz-aminoethyl-NNCA and initiator 2-[(triphenylmethyl)thio]ethanamine were dissolved in anhydrous THF to a final concentration of 1 M and 0.5 M, respectively, as the working solution. The *N*-Cbz protected HS(Naeg)₅, HS(Naeg)₁₀, HS(Naeg)₂₀, and HS(Naeg)₄₀ were synthesized by followed aforementioned protocol from a mixture of Nᵝ-Cbz-aminoethyl-NNCA solution (0.18 mL) and 2-[(triphenylmethyl)thio]ethanamine solution (0.048 mL, 0.024 mL, 0.012 mL, and 0.006 mL, respectively, for 5, 10, 20, and 40

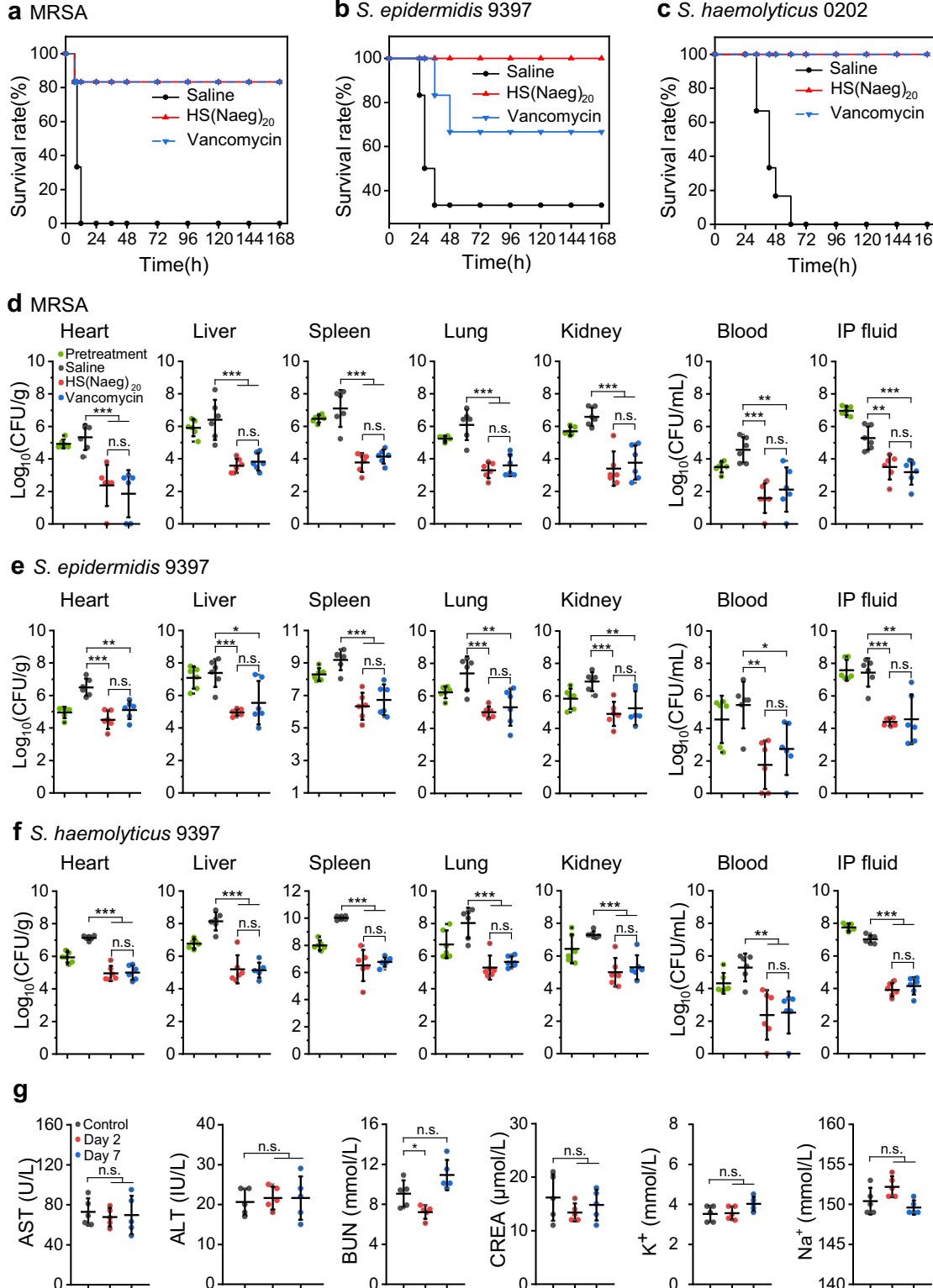

**Fig. 6 HS(Naeg)$_{20}$ displayed in vivo antibacterial efficacy without toxicity in the mouse peritonitis model.** Bacteria suspension was i.p. injected and treatments were administered i.p. at 1 h post infection including, saline, HS(Naeg)$_{20}$ (20 mg/kg), and vancomycin (20 mg/kg). For infections caused by *S. epidermidis* and *S. haemolyticus*, immunosuppressed mice were used. **a–c** Survival rates (7 days) of mice in the peritonitis model induced by MRSA, *S. epidermidis*, and *S. haemolyticus*, respectively. $n = 6$ (six mice in each treatment group in three types of bacterial infection models). **d–f** CFU of bacteria in different organs, blood, and IP fluid in the peritonitis model induced by MRSA, *S. epidermidis*, and *S. haemolyticus*, respectively. $n = 6$ (six mice in each treatment group in three types of bacterial infection models). **g** Blood biochemical analysis at 2 and 7 days post treatment with HS(Naeg)$_{20}$. The untreated mice served as a blank control ($n = 5$ mice per group). Data are presented as mean ± SD. *$P < 0.05$, **$P < 0.01$; ***$P < 0.001$; n.s. (not significant) represents $P > 0.05$ (Student's *t* test).

mer polymer). The *N*-Cbz protected polymers were characterized by GPC using DMF as the mobile phase at a flow rate of 1 mL/min.

**Deprotection of N-Cbz protected poly-Naeg.** *N*-Cbz protected polymers were dissolved in a 1:1 (v/v) mixture of hydrobromic acid in acetic acid (HBr/AcOH, 33 wt. %, 1 mL) and trifluoroacetic acid (TFA, 1 mL) and shaken at rt overnight to remove the protecting group. For polymers using 2-[(triphenylmethyl)thio]etha-namine as the initiator, triethylsilane (40 μL) was used in the reaction of *N*-Cbz deprotection for complete removing of the trityl protecting group. After the solvent was removed under a N$_2$ flow, the reaction mixture was dissolved in 1 mL MeOH, and then cold diethyl ether (45 mL) was added to precipitate out the polymer. The polymer precipitate was collected by centrifugation and removal of the supernatant. After repeating the above dissolution–precipitation process three times, the purified polymer was dissolved in Milli-Q water and lyophilized to obtain a white flocculent solid in the form of bromide salt. The deprotected polymers were characterized by $^1$H NMR.

**Synthesis of N-(3-aminopropyl)-4-morpholin-1,8-naphthalimide (dye–NH$_2$).** *N*-(3-aminopropyl)-4-morpholin-1,8-naphthalimide was synthesized using the reported methods with modifications[67]. 4-Bromo-1,8-naphthalic anhydride (2.8 g, 10 mmol) and morpholine (1.3 g, 15 mmol) was dissolved in methoxylethanol (50 mL) and then the reaction mixture was stirred at 160 °C for 12 h under a nitrogen atmosphere. After cooling to rt, DI water was added to the reaction mixture to precipitate out the intermediate as a yellow solid. The intermediate was dried under vacuum for 12 h, and then mixed with trimethylenediamine (7.4 g, 100 mmol) and 100 mL DI water followed by stirring at 110 °C for 16 h. After removal of most of the solvent, the crude product was purified by silica gel column chromatography to obtain *N*-(3-aminopropyl)-4-morpholin-1,8-naphthalimide as a yellow solid (0.95 g, 28.0% yield over two steps). $^1$H NMR (400 MHz, CDCl$_3$): δ 8.58 (dd, *J* = 1.2, 7.2 Hz, 1H), 8.52 (d, *J* = 8.0 Hz, 1H), 8.42 (dd, *J* = 1.2, 8.4 Hz, 1H), 7.70 (t, *J* = 7.8 Hz, 1H), 7.23 (d, *J* = 8 Hz, 1H), 4.26 (t, *J* = 6.8 Hz, 2H), 4.02–4.00 (m, 4H), 3.27-3.25 (m, 4H), 2.75 (t, *J* = 6.6 Hz, 2H), 1.92-1.86 (m, 4H). HRESI-MS: *m/z* calculated for C$_{19}$H$_{22}$N$_3$O$_3$ [M + H]$^+$: 340.1661; found: 340.1660.

**Synthesis of dye–(Naeg)$_{20}$.** The monomer N$^β$-Cbz-aminoethyl-NNCA and the initiator dye–NH$_2$ were dissolved in anhydrous THF at a concentration of 1 M and 0.5 M, respectively, as the working solution. Dye-conjugated poly-Naeg was synthesized by following the aforementioned protocol from a mixture of N$^β$-Cbz-aminoethyl-NNCA solution (0.18 mL) and dye–NH$_2$ solution (0.012 mL). After the monomer has been consumed, the reaction mixture was directly characterized by GPC using DMF as the mobile phase at a flow rate of 1 mL/min. The polymer was deprotected and purified by following the aforementioned protocol to give the purified dye–(Naeg)$_{20}$ as a yellow solid that is characterized by $^1$H NMR.

**Bacterial strains.** MRSA (*S. aureus* Mu50, *S. aureus* Newman, *S. aureus* USA400, *S. aureus* USA300 LAC, *S. aureus* USA300) were obtained from Shanghai Institute of Materia Medica, Chinese Academy of Sciences. All clinically isolated Gram-positive bacteria used in this study were isolated from patients with various infections, different ages and sexes in the Shanghai Ruijin Rehabilitation Hospital. All strains were cultured in Luria-Bertani (LB) medium at 37 °C before use.

**MIC and MBC assay.** Bacteria were cultured in LB medium at 37 °C for 10 h, and then the bacterial suspension was diluted in Mueller–Hinton (MH) medium to a cell density of $2 \times 10^5$ colony forming units (CFU)/mL as the working suspension. The poly(Naeg)$_{20}$ was serially diluted by MH medium in a 96-well plate, and an equal volume of bacterial suspension was added into each well. The final concentration of the poly(Naeg)$_{20}$ was ranging from 200 to 1.56 μg/mL, and the plates were incubated at 37 °C for 9 h. Wells with inoculum and no polymer were used as the positive control; wells with only MH medium were used as the blank. The optical density (OD) in each well was measured on a Molecular Devices Spectramax M2 precision microplate reader. The percentage of bacteria growth was calculated from

$$\% \text{ cell growth} = \frac{A_{600}^{\text{polymer}} - A_{600}^{\text{blank}}}{A_{600}^{\text{control}} - A_{600}^{\text{blank}}} \times 100 \tag{1}$$

The MIC value was defined as the lowest concentration of an antimicrobial agent to completely inhibit microbial growth. The test was independently repeated at least three times.

The MBC was determined as the lowest concentration of an antimicrobial agent to kill 99.9% of the microbes within the test. For the MBC assay, an aliquot of 3.5 μL mixture from each well of the above MIC test was transferred to a LB agar plate. After incubating the plate at 37 °C for 14 h, the MBC value was determined as the minimum concentration to result in no visible colonies on the plate. The test was independently repeated at least three times.

**Hemolysis assay.** The hemolysis experiment was approved by the Ninth People's Hospital, Shanghai Jiao Tong University School of Medicine. Human blood donors were informed of the hemolysis study and consent was obtained. Human blood was

diluted with Tris-buffered saline (TBS, pH = 7.2) and centrifuged at $1700 \times g$ for 3 min. The human red blood cells (hRBCs) were collected and washed with TBS three times and then diluted to a working concentration of 5% (v/v). The HS(Naeg)$_n$ was serially diluted by TBS in a 96-well plate, and then an equal volume of hRBCs suspension was added into each well. The final concentration of the HS(Naeg)$_n$ was ranging from 10,000 to 78 μg/mL, and the plates were incubated at 37 °C for 1 h. After centrifugation, 80 μL of the supernatant in each well was transferred to a new 96-well plate and the OD value was collected on a microplate reader. Wells containing the mixture of Triton X-100 (3.2 μg/mL in TBS) and hRBCs were used as the positive control; wells containing hRBCs only were used as the blank. The percentage of hemolysis was calculated from

$$\% \text{ hemolysis} = \frac{A_{405}^{\text{polymer}} - A_{405}^{\text{blank}}}{A_{405}^{\text{control}} - A_{405}^{\text{blank}}} \times 100 \tag{2}$$

HC$_{50}$ was defined as the minimum concentration of a compound that causes 50% hemolysis. The test was independently repeated at least three times.

**Cytotoxicity assay.** NIH 3T3 fibroblast cells (ATCC CRL-1658) were cultured in TCPS Petri dishes using Dulbecco's modified eagle medium (DMEM) at 37 °C in the presence of 5% CO$_2$. Subconfluent monolayer culture was trypsinized and then cells were suspended in a growth medium containing serum. After centrifuged at $200 \times g$ for 4 min, cells were collected and resuspended in DMEM and diluted in DMEM to a cell density of $1 \times 10^5$ cell/mL as the working suspension. An aliquot of 100 μL of the cell suspension was added into each well of a 96-well plate and the plate was incubated at 37 °C overnight in presence of 5% CO$_2$. After removing the old medium, fresh DMEM containing HS(Naeg)$_n$ at various concentrations was added to corresponding wells for another 24-h incubation. An aliquot of 10 μL methyl thiazolyl tetrazolium (MTT) solution (5 mg/mL) was added to each well of the plate, and the plate was incubated for 4 h in the dark. After removing the solution, 150 μL of dimethyl sulfoxide (DMSO) was added to dissolve the purple solid. Wells, containing cells and without polymer, were used as the positive control; wells containing DMEM only were used as the blank. The OD values were collected at 570 nm on a microplate reader and the percentage of cell viability was calculated from

$$\% \text{ cell viability} = \frac{A_{570}^{\text{polymer}} - A_{570}^{\text{blank}}}{A_{570}^{\text{control}} - A_{570}^{\text{blank}}} \times 100 \tag{3}$$

to evaluate the cytotoxicity of the HS(Naeg)$_n$.

**Antimicrobial-resistance test.** The antimicrobial-resistance test was conducted according to the previously reported method with slight modification[37]. *S. aureus* ATCC6538 was cultured in LB medium at 37 °C for 10 h and then the bacteria suspension was diluted 400-fold in LB medium containing HS(Naeg)$_{20}$ (0.5 × MBC) or norfloxacin (0.5 × MBC), respectively, followed by incubation at 37 °C under shaking for 24 h. An aliquot of 2.5 μL of the mixture was sampled and diluted 400-fold in LB medium for a new cell-drug incubation cycle. The above cycle was repeated every 24 h. The MBC values of HS(Naeg)$_{20}$ and norfloxacin against *S. aureus* ATCC6538 were examined every 4 days. Based on the MBC value determined by the test, the concentration of polymer and norfloxacin throughout the antimicrobial-resistance study was adjusted to keep at 0.5 × MBC, respectively. Generation of bacteria growth was calculated from incubating time and bacteria growth kinetics described below. An increase in MBC shows the development of drug resistance.

**Bacteria growth Kinetics.** *S. aureus* ATCC6538 was cultured in LB medium at 37 °C for 10 h, and then the bacterial suspension was diluted in MH medium to a cell density of $2 \times 10^6$ CFU/mL as the working suspension. The bacteria suspension was exposed to an equal volume of HS(Naeg)$_{20}$ and norfloxacin solution at a final concentration of 0.5 × MBC at 37 °C. At various time points of treatment, the bacteria samples were diluted for plating on LB agar plate. After incubation at 37 °C for 14 h, the number of colonies on the plate was counted to calculate the rate of bacteria growth.

**Study on bacterial killing kinetics.** *S. aureus* ATCC6538 was cultured in LB medium at 37 °C for 10 h, and then the bacterial suspension was diluted in MH medium to a cell density of $2 \times 10^5$ CFU/mL as the working suspension. The bacteria suspension was treated with an equal volume of HS(Naeg)$_{20}$ or vancomycin solution at final concentrations of 1× and 2 × MBC at 37 °C. At various time points of treatment, the bacteria samples were diluted for plating on LB agar plates. Then the plates were incubated at 37 °C for 14 h for CFU counting and viability calculation. The test was independently repeated at least three times.

**Kill kinetics on persister cell.** *S. aureus* USA300 was cultured in LB medium at 37 °C for 6 h, and then the bacterial suspension was diluted in MH medium to a cell density of $1 \times 10^8$ CFU/mL as the working suspension. 1 mL of bacterial suspension was treated with ciprofloxacin at a concentration of 10 × MIC at 37 °C for 18 h. Half of the bacterial suspension was washed to remove antibiotics and treated with HS(Naeg)$_{20}$ at 4 × MIC at MH medium. The rest bacterial suspension

continued to be treated with ciprofloxacin as the control. At various time points of treatment, the bacteria samples were washed to remove the antibacterial agents and then diluted for plating on LB agar plate. Then the plates were incubated at 37 °C for 14 h for CFU counting and viability calculation.

**Inhibition study on biofilm formation**. S. aureus USA300 was cultured in LB medium at 37 °C for 10 h, and then the bacterial suspension was diluted in MH medium (containing 1% glucose and 1% NaCl) to a cell density of $2 \times 10^5$ CFU/mL as the working suspension. The HS(Naeg)$_{20}$ and antibiotics (norfloxacin and vancomycin) were serially diluted with MH medium (containing 1% glucose and 1% NaCl) in a tissue culture-treated 96-well plate, and then an equal volume of bacterial suspension was added into each well. The plate was incubated at 37 °C for 24 h, and then the bacterial suspension was removed, washed with PBS once, followed by adding 100 μL MTT solution (0.5 mg/mL) into the corresponding well. After further incubation at 37 °C for 4 h in the dark, the MTT solution was removed. An aliquot of 100 μL DMSO was added to each well to solubilize the purple solid and then the OD value was collected at 570 nm on a microplate reader. Wells containing inoculum and without antibacterial agents were used as the positive control; wells containing only MH medium were used as the blank. The percentage of surviving cells within biofilm was calculated from

$$\% \text{ cell viability} = \frac{A_{570}^{\text{polymer}} - A_{570}^{\text{blank}}}{A_{570}^{\text{control}} - A_{570}^{\text{blank}}} \times 100 \tag{4}$$

The test was independently repeated at least three times.

**Activity against mature biofilm**. S. aureus USA300 was cultured in LB medium at 37 °C for 10 h, and then the bacterial cells were inoculated in a tissue culture-treated 96-well plates at initial densities of $10^5$ CFU/mL. The plate was incubated at 37 °C for 24 h to allow biofilm formation. After the old medium was removed from the biofilm plate, fresh MH medium containing various concentrations of HS(Naeg)$_{20}$ and antibiotics (norfloxacin and vancomycin) were added to corresponding wells for another 24 h of incubation. Then MTT assay was used to determine the biofilm viability. Wells containing inoculum and without antibacterial agents were used as the positive control; wells containing only MH medium were used as the blank. The percentage of surviving cells within biofilm was calculated from

$$\% \text{ cell viability} = \frac{A_{570}^{\text{polymer}} - A_{570}^{\text{blank}}}{A_{570}^{\text{control}} - A_{570}^{\text{blank}}} \times 100 \tag{5}$$

The test was independently repeated at least three times.

**Cytoplasmic membrane depolarization assay**. The cytoplasmic membrane depolarization assay was conducted according to the previously reported method with slight modification[68]. S. aureus ATCC6538 was cultured in LB medium at 37 °C for 6 h, and then the bacterial suspension was diluted in HEPES medium ((5 mM HEPES, 20 mM glucose, pH = 7.4) to a cell density of $10^7$ CFU/mL as the working suspension. The bacteria suspension was incubated with 0.8 μM diSC3(5) for 1 h, followed by the addition of KCl to a final concentration of 0.1 M to balance the cytoplasmic and external K$^+$ concentration. An aliquot of 90 μL of the suspension was placed in a 384-well plate and then changes in fluorescence intensity were recorded on a microplate reader (excitation $\lambda = 622$ nm, emission $\lambda = 673$ nm). When the fluorescence intensity remained stable, an aliquot of 10 μL HS(Naeg)$_{20}$ solution was added to a final concentration of $1 \times$ MIC and $2 \times$ MIC, and then the fluorescence intensity was continuously recorded. 0.1% Triton X-100 was used as the positive control.

**Time-lapse fluorescence confocal imaging of bacteria**. S. aureus ATCC6538 was cultured in LB medium at 37 °C for 6 h, and then the bacterial suspension was diluted in RPMI medium to a cell density of $2 \times 10^7$ CFU/mL as the working suspension. An aliquot of 10 μL bacterial suspension was added to a glass-bottom cell culture dish and kept still for 5 min to allow the bacteria to settle and adhere to the bottom. An aliquot of 10 μL RPMI medium containing dye–(Naeg)$_{20}$ ($2 \times$ MBC, green fluorescence) and propidium iodide (20 mM, red fluorescence) was then added to the bacterial drop. Images were taken at the various time points for three channels, the bright field, 562 nm and 488 nm, respectively.

**DNA-binding assay**. DNA-binding assay was conducted using the previously reported method with slight modification[69]. In total, 0.7 μg DNA was mixed with HS(Naeg)$_{20}$ (500 μg/mL in DI water) at different N/P ratios (0.25:1, 0.5:1, 1:1, 2:1, 3:1) and then diluted to 10 μL of solution with DI water. N means the number of amine groups in HS(Naeg)$_{20}$, P means the number of phosphate anions in the plasmid backbone. After incubation for 15 min, the mixtures were analyzed by electrophoresis using 1% agarose gel with ethidium bromide in Tris-borate-EDTA buffer. DNA bands were visualized by using gel documentation and an image analysis system. Native loading buffer containing 10 mM Tris-HCl (pH = 7.5), 40% glycerol, 0.25% bromophenol blue and 0.25% xylene cyanol was used in the test.

**Intracellular ROS assay**. The intracellular ROS levels of S. aureus were determined using the previously reported method with slight modification[70]. S. aureus ATCC6538 was cultured in LB medium at 37 °C for 6 h, and the bacterial suspension was diluted in PBS to a cell density of $2 \times 10^8$ CFU/mL. An aliquot of 5 mL PBS containing DCFH-DA (a reactive oxygen fluorescent probe, 20 μM) was added to an equal volume of bacterial suspension. After incubation for 30 min in the dark, the bacterial cells were washed with PBS twice to remove the DCFH-DA outside the cell and then diluted in PBS to a cell density of $10^8$ CFU/ mL as the working suspension. An aliquot of 90 μL the working suspension was placed in a 384-well plate, then 10 μL of HS(Naeg)$_{20}$ solution ($10 \times$ MBC), PBS, and the mixture of HS(Naeg)$_{20}$ ($10 \times$ MBC) and NAC (100 mM) were added to the bacterial suspension, respectively. The DCF fluorescence intensity was recorded continuously on a microplate reader (excitation $\lambda = 488$ nm, emission $\lambda = 530$ nm). The test was independently repeated twice.

**SEM characterization of bacteria morphology**. S. aureus ATCC6538 was cultured in LB medium at 37 °C for 10 h, and then bacteria suspension was diluted in LB to $2 \times 10^5$ CFU/mL. An aliquot of 500 μL of the HS(Naeg)$_{20}$ solution ($2 \times$ MBC) and an equal volume of the bacterial suspension were mixed in a sterile Eppendorf tube and the mixture was incubated at 37 °C for 30 min. An untreated bacteria suspension was used as the control. HS(Naeg)$_{20}$ treated and untreated bacteria were collected by centrifugation at $1700 \times g$ for 5 min. They were washed with phosphate buffer saline (PBS) once and then fixed with 4% glutaraldehyde in phosphate buffer (PB) at 25 °C overnight. The bacteria were further washed with PBS and dehydrated with gradient ethanol (EtOH) solutions (30, 50, 70, 80, 90, 95, and then 100% ethanol). The samples were dried in air and then used for FESEM characterization.

**TEM characterization of bacteria morphology**. S. aureus ATCC6538 was cultured in LB medium at 37 °C for 10 h, and then the bacterial suspension was diluted in MH medium to a cell density of $10^9$ CFU/mL as the working suspension. An aliquot of 10 mL of the HS(Naeg)$_{20}$ solution ($4 \times$ MBC) and an equal volume of the bacterial suspension were mixed in a sterile centrifugal tube, and the mixture was incubated at 37 °C for 30 min. An untreated bacteria suspension was used as the control. HS(Naeg)$_{20}$ and untreated bacteria were collected by centrifugation at $1700 \times g$ for 5 min. They were washed with PBS once and then fixed with 2.5% glutaraldehyde in PB at 4 °C overnight. After removing the fixed solution, the bacteria cells were rinsed with PB three times and then fixed with 1% osmium acid solution for 1 h. The cells were further washed with PB and dehydrated with gradient ethanol solutions (30, 50, 70, 80, 90, 95, and 100% ethanol). Subsequently, the sample was treated sequentially with acetone for 20 min, the mixture of embedding agent and acetone (v/v = 1/1) for 1 h, the mixture of embedding agent and acetone (v/v = 3/1) for 3 h, and the embedding agent overnight. After osmotic treatment, the samples were embedded and heated overnight at 70 °C. The embedded samples were sectioned in a LEICA EM UC7 ultrathin slicer to obtain 70–90 nm sections. After being stained with lead citrate solution and 50% uranium dioxide-acetate saturated solution for 5 min, the sections were observed under the transmission electron microscope.

**Mouse wound infection model**. All animal procedures were performed in accordance with the Guidelines for Care and Use of Laboratory Animals of the Ninth People's Hospital, Shanghai Jiao Tong University School of Medicine and experiments were approved by the Animal Ethics Committee of the Ninth People's Hospital, Shanghai Jiao Tong University School of Medicine. The laboratory animal usage license number is SYXK-2016-0016, certified by Science and Technology Commission of Shanghai Municipality.

This infection model was conducted according to the previously reported method with slight modification[71]. Female balb/c mice (20–21 g) were used in the full-thickness wound model caused by MRSA. Prior to surgery, mice were anesthetized with an intraperitoneal injection of sodium pentobarbital at a dose of 75 mg/kg. The dorsal hair was shaved, and the skin was rinsed with 75% alcohol three times. The 6-mm-diameter full-thickness wounds were punctured on either side of the back using a biopsy punch. An aliquot of 10 μL S. aureus USA300 LAC suspension ($1.25 \times 10^5$ CFU/mL) was added to wound site. The wound site was covered by Tegaderm dressing (3 M, St Paul, MN) to prevent contamination. At 24 h post infection, mice were randomly divided into three groups (seven animals per group) and then 15 μL of different treatment solutions were administered to the wound site: 0.9% saline (the negative control), HS(Naeg)$_{20}$ (1.56 mg/mL) and vancomycin (1.56 mg/mL, the positive control). The treatments were applied every 4 h for a total of three times. After the last treatment for 4 h, mice were euthanized with an overdose of sodium pentobarbital. The wound sites were excised, weighed, and homogenized in PBS containing 0.1% TX-100. Then the homogenate was serially diluted for plating on LB agar. After incubation at 37 °C for 14 h, the number of colonies on the plate was counted to calculate the bacteria burden of wounds.

For experiments of S. epidermidis and S. haemolyticus infections, female ICR mice (21–23 g) were rendered neutropenic by treatment with 150 and 100 mg/kg of cyclophosphamide intraperitoneally at 4 days and 1 day prior to infection. An

aliquot of 10 µL *S. epidermidis* 9397 suspension ($1.45 \times 10^6$ CFU/mL) or *S. haemolyticus* 0202 suspension ($5 \times 10^5$ CFU/mL) was added to the wound site. At 24 h post infection, mice were randomly divided into three groups (seven mice per group) and then 15 µL of different treatment solutions were administered to the wound site: 0.9% saline (the negative control), HS(Naeg)$_{20}$ (3.13 mg/mL), and vancomycin (3.13 mg/mL, the positive control). The treatments were applied every 4 h for a total of three times. After the last treatment for 4 h, mice were sacrificed and the tissues were harvested to determine the CFU.

**MRSA biofilm formation on contact lens**. The contact lenses were cut into discs with a diameter of 3.5 mm and placed in a 96-well plate. The discs were soaked with 150 µL of MH medium overnight. Then the discs were transferred to a new 96-well plate and 150 µL of *S. aureus* USA300 LAC suspension ($10^5$ CFU/mL in MH medium) was added into corresponding wells. The plate incubated at 37 °C with shaking at 100 rpm. After 2 h incubation, the suspension was aspirated and the lenses were washed with PBS to remove any nonadherent cells, followed by the addition of 150 µL of fresh MH medium. The plate containing the contact lenses with adhered cells was incubated at 37 °C with shaking at 100 rpm for 18 h to allow biofilm formation.

**Mouse keratitis model**. This infection model was conducted according to the previously reported method with modification[26]. Male balb/c mice (22–24 g) were intraperitoneally injected with cyclophosphamide (150 mg/kg at 4 days and 100 mg/kg at 1 day before inducing keratitis) for immunosuppression. Prior to surgery, mice were anesthetized with an intraperitoneal injection of sodium pentobarbital at a dose of 75 mg/kg. In addition, 0.5% tetracaine hydrochloride eye drops were used as topical anesthesia and then a 2 mm-diameter filter paper disc containing 2 µL of 99% 1-heptanol was placed on the center of the cornea for 15 min. The corneal epithelium was scraped off by an iris restorer and the eyes were irrigated with 10 mL of saline to remove any debris and remaining 1-heptanol. A 3.5 mm-diameter contact lens with MRSA biofilm was then placed on the cornea surface. Then the eyelid was closed with 8-0 sutures to keep the contact lens inside. After inoculation for 12 h, the suture was removed, the eyelid was opened and the lens was taken out. The mice with keratitis were randomly divided into three groups (four animals per group): 0.9% saline (the negative control), HS(Naeg)$_{20}$ (1.56 mg/mL), and vancomycin (1.56 mg/mL, the positive control). In total, 10 µL of saline, HS(Naeg)$_{20}$, or vancomycin eye drop was applied to the mice every 5 min during the first hour and every 30 min during the next 7 h. All mice were killed and the eyeballs of each mouse were collected immediately 30 min after the last treatment, followed by the quantitative analysis of bacterial burden. The eyeballs were weighed, and homogenized in PBS containing 0.1% TX-100. Then the homogenate was serially diluted for plating on LB agar. After incubation at 37 °C for 14 h, the number of colonies on the plate was counted to calculate the bacteria burden of each eye.

**Mouse peritonitis model**. This infection model was conducted according to the previously reported method with slight modification[18]. Female ICR mice (21–24 g) were used for MRSA-infected peritonitis model. Mice were infected by intraperitoneal administration of 0.2 mL of *S. aureus* USA300 LAC in saline ($7.5 \times 10^8$ CFU/mL) supplemented with 5% mucin. At 1 h post infection, mice (six animals per group) were intraperitoneally administered with 200 µL aliquot of saline (negative control), HS(Naeg)$_{20}$ (20 mg/kg), or vancomycin (20 mg/kg, positive control). Once the infected mice died, peritoneal lavage was performed by injecting 3.0 mL of saline into the peritoneal cavity and massaging the abdomen. Subsequently, the abdomen was opened and peritoneal fluid was recovered from the abdominal cavity for analysis. Blood samples were collected through cardiac puncture for analysis of CFU/mL. Then, different organs including the heart, liver, spleen, lung, and kidney were removed and homogenized in 0.1% TX-100 solution. The homogenate was serially diluted for plating on LB agar. After incubation at 37 °C for 14 h, the number of colonies on the plate was counted to calculate the bacteria burden. Surviving mice were euthanized at 48 h after infection. The blood, peritoneal fluid, and organs were collected for the determination of bacterial colonies. Besides, the same organs were collected from saline, HS(Naeg)$_{20}$ and vancomycin treated groups for histological analysis. For the pretreatment group, mice were sacrificed and tissues were harvested to determine the CFU at 1 h post infection.

For experiments of *S. epidermidis* and *S. haemolyticus* infections, female ICR mice (22–24 g) were rendered neutropenic by treatment with 150 and 100 mg/kg of cyclophosphamide intraperitoneally at 4 days and 1 day prior to infection. Mice were infected by intraperitoneal administration of 0.2 mL of *S. epidermidis* 9397 ($5.1 \times 10^9$ CFU/mL) or *S. haemolyticus* 0202 ($2.6 \times 10^9$ CFU/mL) in saline. Other experiment conditions were similar to those in MRSA peritonitis model.

For the survival test, the physiological condition of the mice (six animals per group) was observed for 7 days after infection, and the survival rate was recorded.

**In vivo toxicity study**. Female ICR mice (23–25 g) were used to determine the in vivo toxicity of HS(Naeg)$_{20}$. In all, 25 mg/kg of HS(Naeg)$_{20}$ in 200 µL of saline was injected into mice (five animals per group) via tail vein. After 2 and 7 days, blood (0.5–0.8 mL) was collected by cardiac puncture. Clinically significant

biomarkers including alanine transaminase (ALT) level, aspartate transaminase (AST) level, creatinine, urea nitrogen, sodium, and potassium ions levels in blood were analyzed. The untreated group served as a blank control. At 7 days post treatments, the mice were euthanized and liver and kidney were collected for histological analysis.

**Statistics and reproducibility**. Significance between the two groups was determined by two-tailed Student's *t* test. All results were expressed as mean ± standard error. All micrograph assays were carried out at least three independent times with similar results.

**Reporting summary**. Further information on research design is available in the Nature Research Reporting Summary linked to this article.

## Data availability

Data supporting the findings of this study are available within the article and its supplementary materials. Any other data that support the findings of this study are available from the corresponding author, upon reasonable request. Source data are provided with this paper.

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

## Acknowledgements

This research was supported by the National Natural Science Foundation of China for Innovative Research Groups (No. 51621002), the National Natural Science Foundation of China (Nos. 21574038, 21774031, 21861162010), Frontier Science Research Base of Optogenetic Techniques for Cell Metabolism grant 2021 Sci & Tech 03-28 (Shanghai Municipal Education Commission), Program of Shanghai Academic/ Technology Research Leader (20XD1421400), Research Program of State Key Laboratory of Bioreactor Engineering, the Fundamental Research Funds for the Central Universities (JKD 01211520). The authors also thank the Research Center of Analysis and Test of East China University of Science and Technology for the help on the characterization.

## Author contributions

J.X., M.Z., and R.L. designed the experiments, evaluated the data, and wrote the manuscript together; Y.Q., W.J., Z.C., C.D., M.C., Z.J., and J.L. participated in the animal study; W.Z. conducted the time-lapse fluorescent confocal imaging; X.X. and L.L. participated the antimicrobial assays; S.C. conducted the hemolysis assays; N.S. conducted the cytotoxicity assays; J.Z. performed DNA-binding assay; R.L. directed the project.

## Competing interests

R.L. and M.Z. are co-inventors on a patent application covering the reported synthesis of polypeptoids and their antibacterial application. The remaining authors declare no competing interests.
