## [Peer Review File · Nature Communications]

REVIEWER COMMENTS

Reviewer #1 (Remarks to the Author):

This is an interesting paper that demonstrates some interesting molecules, but it is hard to put one's finger onto a single feature that makes it novel. Many papers have been published about such peptoids and polymers and some have shown in vivo efficacy. I think with some major additions the paper may be suitable for further consideration for publication in Nature Communications.

Major Issues

1. The English is understandable but poor and needs considerable copy editing e.g. the title should state "Addressing ..." not Address. Also one cannot have "potent ... potential" (it is either a potent therapeutic or has potential).

2. With respect, species-specific antimicrobials have been discussed previously but at this stage there is not a single one on the market and it is questionable whether such molecules would be economically viable (largely because identification of the specific organism causing infection usually lags behind the need to apply therapies to treat patients). This should at least be addressed but frankly I feel the authors have to at least show this therapy addresses a range of Gram positive infections or even better a range of organisms causing similar infections (respiratory, SSTI etc.).

3. Antibacterial polymers are not new (the authors cite several papers but there are many others). The authors suggest an advantage over HDPs but it must be explicitly stated what is the advantage over existing peptidomimetic anti-bacterial polymers, including a comparison of relative activities. For example in

<https://journals.plos.org/plosone/article/figure?id=10.1371/journal.pone.0073812.t001> the antibacterial polymers have similar but broader spectrum activities to those described here while those in <https://www.nature.com/articles/s41598-019-51561-7/tables/1> are better and broad spectrum.

Other issues

5. Clearly the one pot method used to synthesize these polymers is of interest. Is this a novel feature? If so it should be identified and maybe described in the abstract.

6. Line 67. It was not clear to me how the different lengths of polymers were created.

7. Line 69. The authors should define dispersities for non-Chemists (i.e. heterogeneity of sizes of molecules or particles in a mixture). How was this determined?

8. Line 70. Only 2 pathogens and a total of 3 Gram positive bacteria (2 using single strains) were tested. Other polymers and peptidomimetics have been tested against a far broader range of organisms. A broader spectrum of bacteria should be tested here with some comparison of activities with previous polymers of similar size included in the discussion.

9. Line 73. I would not call these molecules potent. Activities against the target organism are in the 6.25 to 12.5 µg/ml range with MBC up to 25 µg/ml. If the authors feel such activities are potent this should be justified by some comparison. For example Daptomycin has MIC of around 0.2 µg/ml. The resistance to serum and salt are very positive features.

10. Line 102. Can the authors speculate how such a large polymer can enter into cells?

11. Line 109. Every polycation is able to bind to DNA. This is not unusual. Rather that DNA binding somehow generates ROS is surprising and I do not know of a mechanism that would explain this. Can the authors please speculate how this could happen? Nevertheless the fact that a powerful antioxidant NAC blocks killing is indeed interesting. However the papers described show non peptoid agents that damage DNA which has not been shown here, and these papers also do not make a causal link between ROS production and cell killing since the DNA damage itself might be sufficient to cause killing.

12. Figure 5 animal models. It would be useful for the authors to provide brief details about these models in the text or Figure legends. I am a bit concerned with the IP model since peptoid is added within one hour by IP to mice infected IP which looks to me like a mouse test tube experiment.

13. In the wound model have the authors provided any confirmation that the infection is a biofilm infection since the number of bacteria is quite small. In my experience in such models, Staph aureus rapidly declines in numbers and clears without treatment in 2-3 days. I am generally happy

with the keratitis model.

Reviewer #2 (Remarks to the Author):

In this manuscript the authors synthesized a series of host defense peptide mimicking peptoid polymers that have advantages of easy synthesis, low cost and capable of scale up, compared to peptoids obtained from solid phase synthesis. These peptoid polymers display potent activities against MRSA on both planktonic cells, persister cells and biofilms, inhibiting MRSA biofilm formation and eradicating mature biofilms. The authors also did a thorough exploration on the antibacterial mechanism and the finding facilitates further design of antimicrobial agents. Moreover, this work fully demonstrated the therapeutic potential of these peptoid polymers using three animal models, implying the great potential of their structural design and these peptoid polymers as promising antibacterial agents. This manuscript is very well organized with excellent discussion and solid conclusion. I recommend acceptance of this manuscript for publication in Nature Communications after some minor questions below are addressed.

1. What is the definition of n in the peptide polymer structure in Fig1a?
2. In initial study, peptoid polymers with variable C-terminal functional groups were synthesized and studied for their antibacterial activities. The authors should provide comments on this design. Why they choose these terminal functional groups?
3. In Fig2a, test on some extra S.a strains, especially clinically isolated strains, would be helpful to further demonstrate broad spectrum activity of the polymers.
4. Fig3b is too small in its' current form. These confocal images are critical data in revealing the antibacterial mechanism of the peptoid polymer. The author should enlarge this figure and make it more clear and readable.
5. Current scale bar in Fig3h is hard to see.
6. Some necessary reference should be provided in the experimental section. All details should be provided for experimental details such as the replicates for each group.
7. The authors claimed that HS(Naeg)20 had a much faster bacteria killing than normal antibiotics. Although relevant information on the killing kinetics of antibiotics has been reported in literature, a direct comparison can strength this paper.
8. The MIC and MBC of HS(Naeg)20 against S. aureus USA300 and S. aureus USA300 LAC were also measured using MH containing serum and NaCl as the medium. What is the significance of the experiment? What's the purpose of these tests?
9. Pay attention to typos, such as "both plantonic and perssister cells".
10. Terms should be consistent throughout the manuscript such as "CFU mL-1" and "CFU/mL", and "TX100" and "TX-100".

RESPONSE TO REVIEWERS:

Reviewer #1 (Remarks to the Author):

This is an interesting paper that demonstrates some interesting molecules, but it is hard to put one's finger onto a single feature that makes it novel. Many papers have been published about such peptoids and polymers and some have shown in vivo efficacy. I think with some major additions the paper may be suitable for further consideration for publication in Nature Communications.

Response: We thank the reviewer for the favorable comments and detailed suggestions bellow to help us improve our manuscript.

Major Issues

1. *The English is understandable but poor and needs considerable copy editing e.g. the title should state "Addressing ..." not Address. Also one cannot have "potent ... potential" (it is either a potent therapeutic or has potential).*

Response: We thank the reviewer for pointing out this issue. We have followed the suggestion and changed the title to "Addressing MRSA Infection and Antibacterial Resistance with Peptoid Polymers". We also polished our language throughout the manuscript.

2. *With respect, species-specific antimicrobials have been discussed previously but at this stage there is not a single one on the market and it is questionable whether such molecules would be economically viable (largely because identification of the specific organism causing infection usually lags behind the need to apply therapies to treat patients). This should at least be addressed but frankly I feel the authors have to at least show this therapy addresses a range of Gram positive infections or even better a range of organisms casing similar infections (respiratory, SSTI etc.).*

Response: We thank the reviewer for the suggestion that encouraged us to study the antibacterial property of our peptoid polymer on other different Gram-positive strains. We also took a lot effort for *in vivo* evaluation to study the therapeutic potential of the optimal peptoid polymer using more animal infection models. We studied the optimal peptoid polymer against clinically isolated Gram-positive strains, including *Staphylococcus aureus*, *Staphylococcus haemolyticus*, *Staphylococcus epidermidis*, *Enterococcus faecium*, *Enterococcus faecalis*, *Streptococcus agalactiae* and *Listeria monocytogene*. The minimum inhibitory concentrations (MIC) study showed that the optimal peptoid polymer is active against all these Gram-positive bacteria, with MIC values in a range of 1.56-25 µg/mL (Table 1 in our revised manuscript, as shown below).

In the past 6 months, we also built four extra animal models for *in vivo* antibacterial study. In our revision, in addition to the MRSA wound infection model, the MRSA keratitis model and the MRSA peritonitis model in our original manuscript (Figure 5a, 5d, 6a, 6d in our revised manuscript), we also evaluated the antibacterial efficacy of the optimal peptoid polymer using full-thickness wound infection model in mice induced by *S. haemolyticus* or *S. epidermidis*. In this model, the *in vivo* antibacterial efficacy of peptoid polymer is superior to vancomycin

(Figure 5b, 5c in our revised manuscript, as shown below).

We further evaluated the efficacy of the polymer on systemic infections using mouse peritonitis models which were caused by two types of new bacteria. *S. epidermidis* infection caused 66.7% animal death in saline treatment group; whereas, peptoid polymer treatment resulted in 100% survival of the mice (6/6 mice survived), which was better than vancomycin treatment (4/6 mice survived) (Figure 6b in our revised manuscript, as shown below). For the infection caused by *S. hemolyticus*, all saline treated mice died within 60 h; whereas, both peptoid polymer and vancomycin treatment achieved 100% survival of the mice (Figure 6c in our revised manuscript, as shown below). In addition, the peptoid polymer was as effective as vancomycin in reducing the bacterial burden in organs, blood and peritoneal fluid in both two models (Figure 6e, 6f in our revised manuscript, as shown below).

In our revision, we did total seven animal models for *in vivo* antibacterial study, including the mouse full-thickness wound infection models induced by MRSA, *S. epidermidis* and *S. haemolyticus*, a mouse keratitis model induced by MRSA, and the mouse peritonitis models induced by MRSA, *S. epidermidis* and *S. haemolyticus*. These experiments altogether demonstrated the potent *in vivo* anti-infectious efficacy of the optimal peptoid polymer against various Gram-positive bacterial infections.

Table 1. Antibacterial activities against clinically isolated Gram-positive strains

strain	MIC ($\mu\text{g/mL}$)				
	HS(Naeg) ₂₀	Vancomycin	Methicillin	Norfloxacin	Ampicillin
S. aureus 2904	6.25	0.39	6.25	200	25
S. aureus 2802	6.25	0.39	6.25	200	25
S. aureus 2902	6.25	0.39	12.5	25	25
S. aureus 2202	6.25	0.78	3.13	50	1.56
S. haemolyticus 1303	1.56	0.78	>200	25	200
S. haemolyticus 1107	25	0.78	>200	100	>200
S. haemolyticus 0202	3.13	1.56	200	100	200
S. epidermidis 9397	3.13	0.78	1.56	0.2	0.39
S. epidermidis 0501	3.13	0.78	12.5	0.2	6.25
E. faecium 2504	6.25	0.78	>200	200	100
E. faecium 0610	25	0.78	>200	200	>200
E. faecium 1205	25	0.39	>200	100	>200
E. faecium 0502	12.5	>200	>200	100	>200
E. faecium 0609	25	50	>200	>200	>200
E. faecalis 2305	25	0.78	>200	25	12.5
E. faecalis 0609	25	0.39	50	1.56	3.13
S. agalactiae 0613	25	0.78	1.56	25	0.39
L. monocytogenes 1001	12.5	0.39	25	1.56	1.56

Figure 5. HS(Naeg)₂₀ displayed *in vivo* antibacterial efficacy in the mouse full-thickness wound model and the mouse keratitis model. **(a-c)** In the mouse wound model, bacterial suspension was applied to the wound and infected for 24 h followed by topical treatments. For infections caused by *S. epidermidis* and *S. haemolyticus*, immunosuppressed mice were used. **(a)** CFU of MRSA in wound treated with saline, HS(Naeg)₂₀ or vancomycin, all at 1.56 mg/mL. **(b)** CFU of *S. epidermidis* in wound treated with saline, HS(Naeg)₂₀ or vancomycin, all at 3.13 mg/mL. **(c)** CFU of *S. haemolyticus* in wound treated with saline, HS(Naeg)₂₀ or vancomycin, all at 3.13 mg/mL. **(d)** In the mouse keratitis model, contact lens with MRSA biofilm was placed on the injured cornea surface and infected for 12 h followed by topical treatments. CFU of MRSA in cornea treated with saline, HS(Naeg)₂₀ or vancomycin, all at 1.56 mg/mL. * represents $p < 0.05$, ** represents $p < 0.01$; *** represents $p < 0.001$; n.s. (not significant) represents $p > 0.05$ (Student's t test).

Figure 6. HS(Naeg)₂₀ displayed *in vivo* antibacterial efficacy without toxicity in the mouse peritonitis model. Bacteria suspension was i.p. injected and treatments were administered i.p. at 1 h post-infection including Saline, HS(Naeg)₂₀ (20 mg/kg) and vancomycin (20 mg/kg). For infections caused by *S. epidermidis* and *S. haemolyticus*, immunosuppressed mice were used. (a-c) Survival rates (7 d) of mice in the peritonitis model induced by MRSA, *S. epidermidis* and *S. haemolyticus*, respectively. (d-f) CFU of bacteria in different organs, blood and IP fluid in the peritonitis model induced by MRSA, *S. epidermidis* and *S. haemolyticus*, respectively. (g) Blood biochemical analysis at 2 and 7 days post-treatment with HS(Naeg)₂₀. The untreated mice served as a blank control. * represents *p* < 0.05, ** represents *p* < 0.01; *** represents *p* <

0.001; n.s. (not significant) represents $p > 0.05$ (Student's t test).

3. Antibacterial polymers are not new (the authors cite several papers but there are many others). The authors suggest an advantage over HDPs but it must be explicitly stated what is the advantage over existing peptidomimetic anti-bacterial polymers, including a comparison of relative activities. For example in <https://journals.plos.org/plosone/article/figure?id=10.1371/journal.pone.0073812.t001> the antibacterial polymers have similar but broader spectrum activities to those described here while those in <https://www.nature.com/articles/s41598-019-51561-7/tables/1> are better and broad spectrum.

Response: We thank the reviewer for this comment. When we explore new antibacterial agents, we do find polymers with superior antibacterial activity, however, often accompanied with hemolysis or cytotoxicity, which echoed the importance of overall biological performance especially *in vivo* performance, rather than just considering MIC values in finding new antibacterial agents with therapeutic potential. We pay a lot attention to the *in vivo* antibacterial efficacy because we also found polymers displaying excellent antibacterial performance *in vitro* but poor efficacy *in vivo*. As the reviewer mentioned, the poly(oxonorbornene)-based synthetic mimics of antimicrobial peptides (SMAMPs) reported by Prof. Karen Lienkamp (*PLoS ONE* 2013, 8(9), e73812) have similar but broader spectrum activities than our peptoid polymer. However, these SMAMPs have low cell selectivity as shown in the Table of the IC_{50}/MIC_{90} , which is below 10 for 5 out of 7 types of the bacteria in their study. Moreover, these polymers' therapeutic potential is uncertain because the most important *in vivo* antibacterial efficacy has not been studied. The ultra-short triazine based amphipathic polymers (TZP) reported by Prof. Jeong Kyu Bang, Prof. Eun-Kyung Kim and Prof. Song Yub Shin (*Sci. rep.* 2019, 9, 15161) is an excellent work, with high *in vitro* antibacterial activity and cell selectivity. However, the topical application of the TZP was only displayed using atopic dermatitis-like skin lesions in mice caused by 2,4-dinitrochlorobenzene (DNCB) treatment, which is an allergic dermatitis model rather than an infectious model. Due to the lack of *in vivo* anti-infectious efficacy study, the polymers' therapeutic potential is unknown.

Therefore, our effort and contribution is to find promising antibacterial agents that have favorable overall biological activities or therapeutic potential, especially *in vivo* anti-infective performance. Our optimal compound HS(Naeg)₂₀ showed potent *in vivo* anti-infective activity that is superior to vancomycin in three mouse wound infection models and a mouse keratitis model, and as effective as or even slightly better than vancomycin in the mouse peritonitis model for systemic infection. In the meantime, *in vivo* toxicity study indicated that HS(Naeg)₂₀ doesn't show obviously acute toxicity, which reflects the promising prospects of peptoid polymers as therapeutic agents.

As for antibacterial spectrum, actually both broad-spectrum antibacterial agents and narrow spectrum antibacterial agents are highly desired considering the current huge challenge of antimicrobial resistance as said in precedent literatures from Prof. Colin Hill (*Proc. Natl. Acad. Sci.* 2011, 108 (Suppl 1), 4639-44), Prof. Kim Lewis (*Nat. Rev. Drug Discovery* 2013, 12, 371-87). For example, vancomycin is an important antibacterial drug, displaying activity only against Gram-positive bacteria. It is always good to develop an antibacterial agent with

broad-spectrum activity. However, in real application too broad-spectrum antibacterial agents can often kill the normal microorganisms in the human body and may cause unnecessary side effects, such as inducing antimicrobial resistance to Gram-negative bacteria in the treatment on Gram-positive bacterial infection.

We added brief text in our revised manuscript on above discussion that “It’ s worth mentioning that numerous peptidomimetic polymers with antibacterial property have been reported in precedent studies, but most of them were proof-of-concept demonstrations of *in vitro* antibacterial activity and simple *in vivo* studies. The low *in vivo* toxicity and high *in vivo* antibacterial efficacy in multiple animal models demonstrated that our optimal peptoid polymer is a promising candidate for therapeutic agents.”

Other issues

5. Clearly the one pot method used to synthesize these polymers is of interest. Is this a novel feature? If so it should be identified and maybe described in the abstract.

Response: We thank the reviewer for this suggestion and added the description of the one pot method used to synthesize our antibacterial peptoid polymer in the abstract in our revised manuscript that:

“We prepared protease-resistant peptoid polymers with variable C-terminal functional groups using a ring-opening polymerization of *N*-substituted *N*-carboxyanhydrides (NNCA), which can provide peptoid polymers easily from the one-pot synthesis.”

6. Line 67. It was not clear to me how the different lengths of polymers were created.

Response: We thank the reviewer for the question on this important issue, which reminds us to provide a clearer explanation and more details on the synthesis of these peptoid polymers with different chain lengths. The different lengths of peptoid polymers were reacted by using variable ratio of initial monomer/initiator. In our revised manuscript, we made modification that “We prepared thiol terminated poly-Naeg at variable chain lengths by controlling the ratio of initial monomer/initiator (DP = 5, 10, 20, 40; Figure 2b; Figure S15-18, S26-29) and evaluated their activities against *S. aureus* ATCC6538.....”

7. Line 69. The authors should define dispersities for non-Chemists (i.e. heterogeneity of sizes of molecules or particles in a mixture). How was this determined?

Response: We thank the reviewer for this suggestion and added definition (IUPAC recommendations 2009) on dispersity for non-Chemists in our revised manuscript that “dispersity (\mathfrak{D}) is a measure of the dispersion of macromolecular species in a sample of polymer, i.e. a measurement of the heterogeneity of sizes of molecules or particles in a mixture, calculated from the ratio of \bar{M}_w to \bar{M}_n .” The dispersities of the polymers were characterized by GPC using DMF as the mobile phase at a flow rate of 1 mL/min. The method is provided in our revised manuscript (Figure S15-18).

8. Line 70. Only 2 pathogens and a total of 3 Gram positive bacteria (2 using single strains) were tested. Other polymers and peptidomimetics have been tested against a far broader range of organisms. A broader spectrum of bacteria should be tested here with some comparison of activities with previous polymers of similar size included in the discussion.

Response: We thank the reviewer for this suggestion and we have done extra tests on other different Gram-positive bacteria. In our revision, we tested the MIC values of the optimal peptoid polymer against a series of clinically isolated Gram-positive strains, including *Staphylococcus aureus*, *Staphylococcus haemolyticus*, *Staphylococcus epidermidis*, *Enterococcus faecium*, *Enterococcus faecalis*, *Streptococcus agalactiae* and *Listeria monocytogene*. The peptoid polymer is active against these Gram-positive bacteria with MIC values in the range of 1.56-25 µg/mL (Table 1 in our revised manuscript, as shown above in answering question 2 from this reviewer). Comparison with previous polymers as suggested by the reviewer has been discussed in details in our response to the reviewer's 3rd question above. In short, our study takes a lot effort on the extensive tests for *in vivo* antibacterial efficacy and makes a contribution to finding antibacterial polymers with therapeutic potential.

9. Line 73. I would not call these molecules potent. Activities against the target organism are in the 6.25 to 12.5 µg/ml range with MBC up to 25 µg/ml. If the authors feel such activities are potent this should be justified by some comparison. For example Daptomycin has MIC of around 0.2 µg/ml. The resistance to serum and salt are very positive features.

Response: We thank the reviewer for this question and in response to the suggestion we have changed our description from "potent activity" to "effective activity" in our revision. Indeed, as the reviewer said, antibiotics such as daptomycin have very potent antibacterial activity, and their MIC values are often less than 1 µg/mL. However, drug resistance has been emerging continuously and spreading rapidly. In our study, most of the clinically isolated Gram-positive strains were drug-resistant and even multiple drug-resistant (Table 1 in our revised manuscript, as shown above in answering question 2 from this reviewer). Nevertheless, we found that our optimal peptoid polymer exhibited effective antibacterial activity towards all these clinically isolated strains with MIC values in the range of 1.56-25 µg/mL. Moreover, most antibiotics are significantly less effective against bacteria within biofilms. The optimal peptoid polymer eradicated the *S. aureus* USA300 biofilms efficiently at a concentration of 8×MIC; in sharp contrast, vancomycin and norfloxacin couldn't eradicate mature biofilms effectively even at a concentration up to 1024×MIC and 2048×MIC, respectively.

10. Line 102. Can the authors speculate how such a large polymer can enter into cells?

Response: We thank the reviewer for this question. There are a number of examples in literature that cationic antimicrobial agents kill microorganism by acting on internal targets (*Nat. Commun.* 2018, 9, 917; *Expert Rev. Anti Infect. Ther.* 2007, 5(6), 951-959; *J. Med. Chem.* 2018, 61(24), 11101-13).

Our peptoid polymers may be similar to some cationic antimicrobial peptides (Buforin2, *Biochemistry* 2004, 43(49), 15610-16) and probably enter into cytoplasm through direct

translocation, that is, the interaction of the positively charged polymer with negatively charged components of bacteria membrane destabilized the bilayer, creating transmembrane channel, thereby allowing the peptoid polymer to enter the bacterial cells without the membrane lysis. We briefly mentioned above explanation in our revised manuscript that “We hypothesize that peptoid polymer penetrates membrane possibly through direct translocation like some cationic antimicrobial peptides reported in precedent literature: the interaction of the positively charged polymer with negatively charged components of bacteria membrane destabilizes the membrane bilayer, creating transmembrane channel, thereby allowing the peptoid polymer to enter the bacterial cells without the membrane lysis.” with relevant citation.

11. Line 109. Every polycation is able to bind to DNA. This is not unusual. Rather that DNA binding somehow generates ROS is surprizing and I do not know of a mechanism that would explain this. Can the authors please speculate how this could happen? Nevertheless the fact that a powerful antioxidant NAC blocks killing is indeed interesting. However the papers described show non peptoid agents that damage DNA which has not been shown here, and these papers also do not make a causal link between ROS production and cell killing since the DNA damage itself might be sufficient to cause killing.

Response: We thank the reviewer for this question that reminds us to further clarify the mechanism hypothesis. Previous studies (*Proc. Natl. Acad. Sci.* 2019, 116 (20), 10064-71; *Proc. Natl. Acad. Sci.* 2014, 111 (20), E2100-09; *Cell* 2007, 130, 797-810) indicated that interactions of drug with DNA can trigger stress responses that induce redox-related physiological alterations, such as altered metabolism and respiration, thereby resulting in the formation of ROS which further contribute to bacterial cell death.

Precedent literatures indicate that DNA damage itself may not be sufficient to kill bacteria. Prof. James Collins' research (*Proc. Natl. Acad. Sci.* 2014, 111 (20), E2100-09) found that, for DNA-targeting antibiotics, “overexpression of catalase or DNA mismatch repair enzyme, MutS, and antioxidant pretreatment limit antibiotic lethality, indicating that ROS causatively contribute to antibiotic killing.” Similarly, the study by Prof. Xilin Zhao (*Proc. Natl. Acad. Sci.* 2019, 116 (20), 10064-71) showed that the ability of ROS-mitigating chemicals prevent bacterial cell death after removing the DNA damaging agents. These studies indicate that primary DNA damage may not be sufficient to kill bacteria.

In addition, we also did extra experiments to compared the time-kill kinetics of HS(Naeg)₂₀ and DNA-targeting quinolone antibiotics, including norfloxacin, ciprofloxacin and levofloxacin. As shown in the figure below (Figure S40 in our revised manuscript), our peptoid polymer HS(Naeg)₂₀ killed 90% of *S. aureus* within 5 min at a concentration of 2×MBC. In sharp contrast, it takes 2-3 hours for quinolone antibiotics (norfloxacin, ciprofloxacin and levofloxacin) to kill 90% of *S. aureus* at a concentration of 2×MBC. Furthermore, other DNA-targeting antibacterial agents, such as gepotidacin, bis-benzimidazoles, and analogues of distamycin A, are reported to require hours and even longer to take effect (*J. Med. Chem.* 2004, 47, 4352-55; *Biochemistry* 2012, 51, 29, 5860-71; *Antimicrob. Agents Chemother.* 2017, 61(7), e00468-17).

The very fast killing of peptoid polymer than do DNA-targeting antibacterial agents against *S. aureus* and the fact that the antioxidant reagent NAC (Figure 3g in our revised

manuscript, as shown below) blocks cell killing suggest that the DNA damage caused by peptoid may be insufficient, and ROS likely plays a key role in bacterial killing for our peptoid polymer. We also briefly mentioned above discussion in our revised manuscript to clarify our hypothesis on antibacterial mechanism that

“The fact that the antioxidant reagent NAC blocks bacterial killing and that peptoid polymer kills bacteria much faster than do DNA-targeting antibacterial agents (Figure S40) suggest that peptoid polymer kills bacterial dominantly via the generation of high level of ROS, rather than just interaction or damage on DNA. These studies suggested the probable antibacterial mechanism of poly-Naeg: the polymer entered into bacteria and interacted with DNA to trigger stress responses that induce redox-related physiological alterations, thereby, resulting in the generation of high level of ROS and killing bacteria by damaging bacterial membrane.....”

Figure S40. Bacterial killing kinetics of HS(Naeg)₂₀, norfloxacin, ciprofloxacin and levofloxacin against *S. aureus* ATCC6538 at a concentration of 2×MBC.

Figure 3. (g) The MBC of HS(Naeg)₂₀ against *S. aureus* in the presence or absence of NAC (10 mM).

12. Figure 5 animal models. It would be useful for the authors to provide brief details about these models in the text or Figure legends. I am a bit concerned with the IP model since peptoid is added within one hour by IP to mice infected IP which looks to me like a mouse test tube experiment.

Response: We thank the reviewer for this suggestion and we have provided brief details about animal model in the figure legends in our revised manuscript as shown below:

“**Figure 5.** HS(Naeg)₂₀ displayed *in vivo* antibacterial efficacy in the mouse full-thickness wound model and the mouse keratitis model. (a-c) In the mouse wound model, bacterial suspension was applied to the wound and infected for 24 h followed by topical treatments. For infections caused by *S. epidermidis* and *S. haemolyticus*, immunosuppressed mice were used.....(d) In the mouse keratitis model, contact lens with MRSA biofilm was placed on the injured cornea surface and infected for 12 h followed by topical treatments.....”

“**Figure 6.** HS(Naeg)₂₀ displayed *in vivo* antibacterial efficacy without toxicity in the mouse peritonitis model. Bacteria suspension was i.p. injected and treatments were administered i.p. at 1 h post-infection including Saline, HS(Naeg)₂₀ (20 mg/kg) and vancomycin (20 mg/kg). For infections caused by *S. epidermidis* and *S. haemolyticus*, immunosuppressed mice were used.....”

In the mouse peritonitis model, we followed the protocol in literature (*Nat. Microbiol.* 2020, 5, 1040-50; *Proc. Natl. Acad. Sci.* 2019, 116 (52), 26516-22; *Nat. Commun.* 2018, 9, 917; *Angew. Chem. Int. Ed.* 2017, 56, 1486-90) to start treatment at 1 h post-infection. Besides, we also determined pretreatment bacterial cell counts at 1 h post-infection. Obviously, the bacteria rapidly spread to major organs, blood and peritoneal fluid and then caused systemic infections on mice in all three peritonitis models (Figure 6d-6f in our revised manuscript, as shown below).

Figure 6. HS(Naeg)₂₀ displayed *in vivo* antibacterial efficacy without toxicity in the mouse peritonitis model. Bacteria suspension was i.p. injected and treatments were administered i.p. at 1 h post-infection including Saline, HS(Naeg)₂₀ (20 mg/kg) and vancomycin (20 mg/kg). For infections caused by *S. epidermidis* and *S. haemolyticus*, immunosuppressed mice were used. (d-f) CFU of bacteria in different organs, blood and IP fluid in the peritonitis model induced by MRSA, *S. epidermidis* and *S. haemolyticus*, respectively. * represents $p < 0.05$, ** represents

$p < 0.01$; *** represents $p < 0.001$; n.s. (not significant) represents $p > 0.05$ (Student's t test).

13. In the wound model have the authors provided any confirmation that the infection is a biofilm infection since the number of bacteria is quite small. In my experience in such models, *Staph aureus* rapidly declines in numbers and clears without treatment in 2-3 days. I am generally happy with the keratitis model.

Response: We thank the reviewer for the comment. In mouse wound infection model, we followed the protocol in literature (by Prof. Mary B. Chan-Park et al., *Nano Lett.* 2018, 18, 4180–87; *ACS Infect. Dis.* 2020, 6, 1228–37), where it's claimed as a biofilm infection. Nevertheless, we took the reviewer's suggestion not to claim it as biofilm infection without direct proof and made modification in our revised manuscript that "In the wound infection model, MRSA suspension was applied to the wound and infected for 24 h followed by topical treatments with peptoid polymer."

Besides, we also did two full-thickness wound infection models induced by *S. haemolyticus* and *S. epidermidis*, respectively, in neutropenic mice to evaluate the antibacterial efficacy of the peptoid polymer. In these two models, the polymer also significantly reduced the bacteria density on the wound and performed comparable to or even superior to vancomycin (Figure 5b, 5c in our revised manuscript, as shown below).

Figure 5. HS(Naeg)₂₀ displayed *in vivo* antibacterial efficacy in the mouse full-thickness wound model and the mouse keratitis model. (b-c) In the mouse wound model, bacterial suspension was applied to the wound and infected for 24 h followed by topical treatments. For infections caused by *S. epidermidis* and *S. haemolyticus*, immunosuppressed mice were used. (b) CFU of *S. epidermidis* in wound treated with saline, HS(Naeg)₂₀ or vancomycin, all at 3.13 mg/mL. (c) CFU of *S. haemolyticus* in wound treated with saline, HS(Naeg)₂₀ or vancomycin, all at 3.13 mg/mL. * represents $p < 0.05$, ** represents $p < 0.01$; *** represents $p < 0.001$; n.s. (not significant) represents $p > 0.05$ (Student's t test).

Reviewer #2 (Remarks to the Author):

In this manuscript the authors synthesized a series of host defense peptide mimicking peptoid polymers that have advantages of easy synthesis, low cost and capable of scale up, compared to peptoids obtained from solid phase synthesis. These peptoid polymers display potent activities against MRSA on both planktonic cells, persister cells and biofilms, inhibiting MRSA

biofilm formation and eradicating mature biofilms. The authors also did a thorough exploration on the antibacterial mechanism and the finding facilitates further design of antimicrobial agents. Moreover, this work fully demonstrated the therapeutic potential of these peptoid polymers using three animal models, implying the great potential of their structural design and these peptoid polymers as promising antibacterial agents. This manuscript is very well organized with excellent discussion and solid conclusion. I recommend acceptance of this manuscript for publication in *Nature Communications* after some minor questions below are addressed.

Response: We thank the reviewer for the favorable comments and questions below.

1. What is the definition of n in the peptide polymer structure in Fig1a?

Response: We thank the reviewer for the question on this very important issue, which reminds us providing extra explanation in our revised manuscript that “..... n represents the average number of repeating units within the peptoid polymer chain, that is, the degree of polymerization.”

2. In initial study, peptoid polymers with variable C-terminal functional groups were synthesized and studied for their antibacterial activities. The authors should provide comments on this design. Why they choose these terminal functional groups?

Response: We thank the reviewer for the question. We designed a series of peptoid polymers with variable C-terminal functional groups because terminal functional groups are reported to affect the biological function of antimicrobial polymers (*J. Am. Chem. Soc.* 2009, 131(28), 9735-45). We take the reviewer’s suggestion and added brief comments on our design in our revision that "Previous study indicated that biological function of antimicrobial polymers can be tuned by their terminal functional groups, which inspired us to explore peptoid polymers with hydrophilic group PEG₄ or hydrophobic groups, variable aromatic groups and variable alkyl chains."

3. In Fig2a, test on some extra *S.a* strains, especially clinically isolated strains, would be helpful to further demonstrate broad spectrum activity of the polymers.

Response: We thank the reviewer for this suggestion and we have done extra tests on other different clinically isolated bacteria. Table 1 showed the minimum inhibitory concentrations of the optimal peptoid polymer against a series of clinically isolated Gram-positive strains, including *Staphylococcus aureus*, *Staphylococcus haemolyticus*, *Staphylococcus epidermidis*, *Enterococcus faecium*, *Enterococcus faecalis*, *Streptococcus agalactiae* and *Listeria monocytogene*. The peptoid polymer is active against these Gram-positive bacteria with MIC values in the range of 1.56-25 $\mu\text{g/mL}$. In particular, the MIC values of the polymer against the four MRSA strains are 6.25 $\mu\text{g/mL}$ (Table 1 in our revised manuscript, as shown below).

Table 1. Antibacterial activities against clinically isolated Gram-positive strains

strain	MIC ($\mu\text{g/mL}$)				
	HS(Naeg) ₂₀	Vancomycin	Methicillin	Norfloxacin	Ampicillin
S. aureus 2904	6.25	0.39	6.25	200	25
S. aureus 2802	6.25	0.39	6.25	200	25

S. aureus 2902	6.25	0.39	12.5	25	25
S. aureus 2202	6.25	0.78	3.13	50	1.56
S. haemolyticus 1303	1.56	0.78	>200	25	200
S. haemolyticus 1107	25	0.78	>200	100	>200
S. haemolyticus 0202	3.13	1.56	200	100	200
S. epidermidis 9397	3.13	0.78	1.56	0.2	0.39
S. epidermidis 0501	3.13	0.78	12.5	0.2	6.25
E. faecium 2504	6.25	0.78	>200	200	100
E. faecium 0610	25	0.78	>200	200	>200
E. faecium 1205	25	0.39	>200	100	>200
E. faecium 0502	12.5	>200	>200	100	>200
E. faecium 0609	25	50	>200	>200	>200
E. faecalis 2305	25	0.78	>200	25	12.5
E. faecalis 0609	25	0.39	50	1.56	3.13
S. agalactiae 0613	25	0.78	1.56	25	0.39
L. monocytogenes 1001	12.5	0.39	25	1.56	1.56

4. Fig3b is too small in its' current form. These confocal images are critical data in revealing the antibacterial mechanism of the peptoid polymer. The author should enlarge this figure and make it more clear and readable.

Response: We thank the reviewer for this suggestion and we have enlarged Figure 3b in our revision to make it clear and readable, as shown below.

5. Current scale bar in Fig3h is hard to see.

Response: We thank the reviewer for reminding us on this. We have redrawn the scale bar in Figure 3h in our revision to make it clear, as shown below.

Figure 3. Antibacterial mechanism study of poly-Naeg. **(a)** Cytoplasmic membrane depolarization by HS(Naeg)₂₀ at 1×MIC and 2×MIC concentration. **(b)** Time-laps confocal fluorescence imaging on the interaction between *S. aureus* and Dye-(Naeg)₂₀ at a concentration of 1×MBC, in the presence of PI. **(c)** Fluorescence intensity versus time in green and red channels in ROI. **(d)** Ortho view of Z-stack images in (b). **(e)** The electrophoretic mobility shift assay of plasmid DNA and the mixture of plasmid DNA-complexes at different ratios of N:P (HS(Naeg)₂₀:DNA). **(f)** Fluorescence intensity produced by *S. aureus* treated with PBS buffer, HS(Naeg)₂₀ (1×MBC) and the combination of HS(Naeg)₂₀ (1×MBC) and NAC (10 mM), in the presence of 2',7'-dichlorofluorescein diacetate, *** indicated $p < 0.001$ (Student's t test). **(g)** The MBC of HS(Naeg)₂₀ against *S. aureus* in the presence or absence of NAC (10 mM). **(h)** **(i)** SEM and TEM characterization respectively on *S. aureus* cells with and without HS(Naeg)₂₀ treatment at concentration of 1×MBC.

6. Some necessary reference should be provided in the experimental section. All details should be provided for experimental details such as the replicates for each group.

Response: We thank the reviewer for this suggestion. We have supplemented necessary references (*Nano Lett.* 2018, 18, 4180–87; *Nat. Commun.* 2013, 4, 2861; *Angew. Chem. Int. Ed.* 2017, 56, 1486-90) in the method section and provided the details about the replicates for each group in our revised manuscript, for example:

“At 24 h post-infection, mice were randomly divided into three groups (7 animals per group) and then 15 μ L of different treatment solutions were administered to the wound site.....”

In addition, we provided brief experimental details in figure legends:

“**Figure 5.** HS(Naeg)₂₀ displayed *in vivo* antibacterial efficacy in the mouse full-thickness wound model and the mouse keratitis model. (a-c) In the mouse wound model, bacterial suspension was applied to the wound and infected for 24 h followed by topical treatments. For infections caused by *S. epidermidis* and *S. haemolyticus*, immunosuppressed mice were used.....(d) In the mouse keratitis model, contact lens with MRSA biofilm was placed on the injured cornea surface and infected for 12 h followed by topical treatments.....”

“**Figure 6.** HS(Naeg)₂₀ displayed *in vivo* antibacterial efficacy without toxicity in the mouse peritonitis model. Bacteria suspension was i.p. injected and treatments were administered i.p. at 1 h post-infection including Saline, HS(Naeg)₂₀ (20 mg/kg) and vancomycin (20 mg/kg). For infections caused by *S. epidermidis* and *S. haemolyticus*, immunosuppressed mice were used.....”

7. The authors claimed that HS(Naeg)₂₀ had a much faster bacteria killing than normal antibiotics. Although relevant information on the killing kinetics of antibiotics has been reported in literature, a direct comparison can strength this paper.

Response: We thank the reviewer for this suggestion and we have supplemented the bacterial killing kinetics of vancomycin on *S. aureus* to compare the killing rate of peptoid polymer and vancomycin. The time-killing kinetic study showed that HS(Naeg)₂₀ achieved about 2.7-log reduction of *S. aureus* within 60 min at a concentration of 1 \times MBC. In sharp contrast, vancomycin caused only about 0.3-log reduction of the bacteria even after 4 hours of treatment. Compared with conventional antibiotics, the fast bacterial killing is one of the advantages of our peptoid polymers, which is very important in the treatment of sepsis and other situations where there is an urgent need to kill bacteria as quick as possible (Figure 4b in our revised manuscript, as shown below). A brief comment on this study was also added to our revised manuscript.

Figure 4. (b)Bacterial killing kinetics of HS(Naeg)₂₀ and vancomycin against *S. aureus*

ATCC6538 at 1×MBC and 2×MBC concentration.

8. The MIC and MBC of HS(Naeg)20 against *S. aureus* USA300 and *S. aureus* USA300 LAC were also measured using MH containing serum and NaCl as the medium. What is the significance of the experiment? What's the purpose of these tests?

Response: We thank the reviewer for the question. We test the antibacterial efficacy of peptoid polymer by adding 10% fetal bovine serum (FBS) to MH medium to simulate the physiological environment as reported previously in literature from multiple labs such as by Prof. Bob Hancock (*Sci. Rep.* 2017, 7:43321) and by Prof. Yi Yan Yang (*Biomaterials*, 2017, 127, 36-48). As Prof. Guangshun Wang (*Proc. Natl. Acad. Sci.* 2019, 116 (27), 13517-22) and Prof. Jianjun Cheng did (*Proc. Natl. Acad. Sci.* 2015, 112 (43), 13155-60) in their studies, we also tested the effect of physiologically relevant salts (1 mM Mg²⁺, 2 mM Ca²⁺ and 150-200 mM NaCl) in the medium and found no change in the MIC and MBC values of our peptoid polymer against two representative *S. aureus* using MH that contains MgCl₂, CaCl₂ or NaCl (Table S1 in our revised manuscript, as shown below).

Table S1. MIC and MBC values of HS(Naeg)₂₀ in the presence of physiologically relevant salts against *S. aureus* USA300 and *S. aureus* USA300 LAC.

strain	MIC/MBC (µg mL ⁻¹)				
	No salts added	1 mM MgCl ₂	2 mM CaCl ₂	150 mM NaCl	200 mM NaCl
S. aureus USA300	12.5/12.5	12.5/12.5	12.5/12.5	12.5/12.5	12.5/12.5
S. aureus USA300 LAC	6.25/6.25	6.25/6.25	6.25/6.25	6.25/6.25	6.25/6.25

9. Pay attention to typos, such as “both planktonic and persister cells”.

Response: We thank the reviewer for reminding us on this. We have corrected the typos throughout our manuscript.

10. Terms should be consistent throughout the manuscript such as “CFU mL⁻¹” and “CFU/mL”, and “TX100” and “TX-100”.

Response: We thank the reviewer for pointing out this. We have changed the term “CFU mL⁻¹” to “CFU/mL” and “TX100” to “TX-100” throughout the manuscript in our revision to make the terms consistent.

We greatly thank all the reviewers' valuable comments, which help us substantially improved our manuscript. We hope that the revised manuscript will prove to be acceptable for publication in *Nature Communications*.

Sincerely,

Runhui Liu

Professor of Chemistry and Biomaterials

REVIEWERS' COMMENTS

Reviewer #1 (Remarks to the Author):

I have now read all of the responses and I'm generally happy except for the one about ROS generation being the mechanism of action. In fact the Collins work was refuted by 2 papers in Nature and one in Science. I'm ok if the author's suggest that the mechanism might be quite complex (i.e. multihit as discussed by Hancock and Sahl. 2006. Nature Biotech. 24:1551-7), including action on DNA and ROS generation.

Reviewer #2 (Remarks to the Author):

The authors has done an excellent job in addressing reviewers' concerns and has addressed all my concerns. I suggest acceptance of the manuscript.

Point-by-point response to the reviewers' comments

REVIEWER COMMENTS

Reviewer #1 (Remarks to the Author):

I have now read all of the responses and I'm generally happy except for the one about ROS generation being the mechanism of action. In fact the Collins work was refuted by 2 papers in Nature and one in Science. I'm ok if the author's suggest that the mechanism might be quite complex (i.e. multihit as discussed by Hancock and Sahl. 2006. Nature Biotech. 24:1551-7), including action on DNA and ROS generation.

Response: We thank the reviewer for pointing out this and previous suggestions to help us improve our manuscript. Indeed, we can't rule out other factors in bacterial killing, such as, the strong binding of peptoid polymer to DNA may cause the death of bacteria by inhibiting cellular functions. Therefore, we agree with the reviewer's suggestion and modified our mechanism statement to "...the probable complex antibacterial mechanism of poly-Naeg including the generation of high level of ROS and DNA binding" in our revised manuscript.

Reviewer #2 (Remarks to the Author):

The authors has done an excellent job in addressing reviewers' concerns and has addressed all my concerns. I suggest acceptance of the manuscript.

Response: We thank the reviewer for the positive response and suggestions to help us improve our manuscript.

We greatly appreciate reviewers' valuable comments. We hope that the revised manuscript will prove to be acceptable for publication in *Nature Communications*.

Sincerely,

Runhui Liu

Professor of Chemistry and Biomaterials